# TERRA-LSD1 phase separation promotes R-loop formation for telomere maintenance in ALT cancer cells

Meng Xu[1], Dulmi Senanayaka[2], Rongwei Zhao[1], Tafadzwa Chigumira[1], Astha Tripathi[1], Jason Tones[1], Rachel M. Lackner[3], Anne R. Wondisford[4], Laurel N. Moneysmith[2], Alexander Hirschi[5], Sara Craig[2], Sahar Alishiri[2], Roderick J. O'Sullivan ®[4], David M. Chenoweth[3], Nicholas J. Reiter[2] & Huaiying Zhang ®[1] ✉

The telomere repeat-containing RNA (TERRA) forms R-loops to promote homology-directed DNA synthesis in the alternative lengthening of telomere (ALT) pathway. Here we report that TERRA contributes to ALT via interacting with the lysine-specific demethylase 1A (LSD1 or KDM1A). We show that LSD1 localizes to ALT telomeres in a TERRA dependent manner and LSD1 function in ALT is largely independent of its demethylase activity. Instead, LSD1 promotes TERRA recruitment to ALT telomeres via RNA binding. In addition, LSD1 and TERRA undergo phase separation, driven by interactions between the RNA binding properties of LSD1 and the G-quadruplex structure of TERRA. Importantly, the formation of TERRA-LSD1 condensates enriches the R-loop stimulating protein Rad51AP1 and increases TERRA-containing R-loops at telomeres. Our findings suggest that LSD1-TERRA phase separation enhances the function of R-loop regulatory molecules for ALT telomere maintenance, providing a mechanism for how the biophysical properties of histone modification enzyme-RNA interactions impact chromatin function.

Chromatin-associated RNAs play important regulatory roles in chromatin architecture[1]. These RNAs can hybridize with DNA to form DNA–RNA hybrid structures called R-loops. R-loop formation may serve functional roles, such as pausing RNA polymerase II to repress transcription, but can also be detrimental to genome stability by creating barriers for replication machinery[2]. In addition, chromatin-associated RNAs can also scaffold chromatin modifiers and transcriptional factors, including the polycomb repressive complex 2 (PRC2) and the lysine-specific demethylase 1A (LSD1), to regulate chromatin function[3–5].

The telomere repeat-containing RNA (TERRA) is a long noncoding RNA transcribed by RNA Polymerase II from a subset of telomeres[6]. Transcribed TERRA can localize to other telomeres through hybridizing with telomere DNA to form R-loops[7–9]. TERRA also interacts with a wide range of proteins, including histone-associated protein HP1[10], methylation regulatory enzymes such as yeast disruptor of telomeric silencing-1 (DOT1), human methyltransferase 3 (METTL3), LSD1, PRC2[7,11–14], and DNA repair factors such as RNA-binding protein fused in sarcoma (FUS) and breast cancer type 1 susceptibility protein (BRCA1)[15,16]. Through forming R-loop and interacting with various proteins, TERRA plays several surveillance roles that include facilitating telomere DNA replication and protecting stressed/shortened telomeres[7,17–19].

[1]Department of Biology, Carnegie Mellon University, Pittsburgh, PA 15213, USA. [2]Klingler College of Arts and Sciences, Department of Chemistry, Marquette University, Milwaukee, WI 53233, USA. [3]Department of Chemistry, University of Pennsylvania, Philadelphia, PA 19014, USA. [4]Department of Pharmacology and Chemical Biology, UPMC Hillman Cancer Center, University of Pittsburgh, Pittsburgh, PA 15213, USA. [5]Cepheid Diagnostics, 904 E. Caribbean Dr., Sunnyvale, California 94089, USA. ✉e-mail: huaiyinz@andrew.cmu.edu

TERRA is upregulated in cancer cells that use the alternative lengthening of telomere (ALT) pathway for telomere maintenance. ALT cancer cells lack telomerase and instead use homology-directed DNA synthesis to counteract replication induced telomere shortening. This requires many DNA repair and recombination proteins to be localized to ALT telomeres[20]. In addition, ALT cells are characterized by several hallmarks, including the presence of ALT telomere associated PML bodies (APBs), telomere clustering, and upregulation of TERRA[6,20,21]. Although ALT phenotypes are well-characterized, the molecular mechanisms underlying the ALT pathway are still poorly understood.

Upregulated TERRA in ALT cancer cells forms R-loops to promote DNA damage response for homology-directed telomere DNA repair[8,21]. Recent studies show that TERRA also assists DNA synthesis by forming DR-loops, the homologous recombination intermediates that contain both DNA–DNA and DNA–RNA hybrids[22–24]. Depletion of the R-loop by overexpressing RNase H1 impairs the recombinogenic nature of ALT telomeres, leading to telomere shortening[21]. However, whether and how TERRA acts as a scaffold for proteins in the ALT pathway remains largely unknown.

In this work, we screened for proteins that rely on TERRA for localization to ALT telomeres. We find that lysine-specific demethylase 1A (LSD1), also known as KDM1A, associates with TERRA as part of its localization to ALT telomeres. Previous in vitro work has shown that LSD1 interacts with TERRA with nM affinity in a manner that depends on G-quadruplex formation of TERRA[13,25]. In addition, G-quadruplex RNA binding acts as a non-competitive inhibitor of LSD1 demethylation activity[25]. In human cells, LSD1-TERRA interaction is observed on uncapped telomeres to stimulate MRE11 nuclease activity in a manner that is independent of LSD1's demethylation activity[26]. In this study, we find that LSD1 plays a role in ALT telomere maintenance by interacting with TERRA largely independent of its enzymatic activity. In addition, TERRA drives LSD1 to undergo phase separation in length- and RNA structure-dependent manner. We provide evidence that LSD1 condensates enrich TERRA and R-loop stimulating proteins on ALT telomeres, leading to an increase in R-loops. These results uncover a pathway for R-loop formation that involves TERRA-mediated LSD1 phase separation, supporting a regulatory mechanism for RNA-protein interaction networks at ALT telomeres.

## Results

### TERRA is required for LSD1 localization to ALT telomeres
To determine whether TERRA regulates ALT telomere maintenance through a protein interaction network, we first depleted TERRA to confirm its effect on ALT phenotypes. We adopted a published method using single-stranded antisense oligonucleotides (ASO), which harbor locked nucleic acid (LNA) chemistry and a gapmer designed to promote RNaseH-mediated degradation of TERRA RNA in vivo[7]. This approach efficiently depleted TERRA, as demonstrated by TERRA northern-dot blot, fluorescence situ hybridization (FISH), and qRT-PCR (Fig. 1a, b and Supplementary Fig. 1a, b). In ALT positive U2OS cells synchronized in the G2 stage, where the ALT pathway is active, we observed decreased telomere clustering after TERRA depletion (Supplementary Fig. 1c, d). This telomere clustering is a unique ALT feature that is known to facilitate homology-directed telomere synthesis[27,28] and is evaluated by quantifying the number of large foci in telomere DNA FISH images (for details see Methods).

To confirm the role of TERRA in ALT, we also employed the well-established TRF1-FokI system[28]. In this system, the expression of the FokI nuclease fused to telomere repeat factor 1 (TRF1) triggers damage exclusively within the telomeric DNA, leading to strong ALT features, such as telomere clustering, the formation of ALT-associated promyelocytic leukemia nuclear bodies (APBs), and POLD3-dependent telomere DNA synthesis in non-S cells. In addition, this system allows us to induce ALT phenotypes with the addition of a small molecule

4-Hydroxyestradiol (4-OH) to translocate FokI into the nucleus, facilitating a direct comparison of ALT phenotypes under different conditions. Consistent with the results in G2 cells, TERRA depletion also reduces FokI-induced telomere clustering, whereas only a modest decrease was seen in the inactive FokI mutant FokI[D450A] control cells (Fig. 1c, d). Furthermore, APBs are significantly reduced in TERRA knockdown cells (Fig. 1e, f) and newly synthesized telomeric DNAs, which are pulsed with fluorescent Edu (5-Ethynyl-2-deoxyuridine), are decreased after TERRA depletion (Fig. 1g, h). Taken together, these data confirm that ASO can effectively knock down TERRA to evaluate its functions in ALT activity.

With siTERRA, we asked which RNA interacting proteins localize to ALT telomeres and whether protein localization depends on TERRA. We expressed fluorescently labeled RNA interacting proteins, including NONO, hnRNPUL1, FUS, RBXP, and LSD1, in TRF1-FokI cells with siCtr and siTERRA. For EZH2, we used immunostaining because of the availability of a good antibody. While most proteins showed little localization to telomeres, LSD1 and EZH2 were enriched at telomeres in siCtr cells and their localization is reduced in siTERRA cells (Fig. 1i, j, Supplementary Fig. 2a–d). This suggests that LSD1 and EZH2 are involved in ALT in a TERRA-dependent manner. We subsequently focused on LSD1 for further investigation because it demonstrated the most pronounced reduction after siTERRA treatment.

To examine whether LSD1 telomeric localization is ALT activity dependent, we compared LSD1 localization in FokI and nuclease dead FokI[D450A] mutant cells. Here, reduced localization of LSD1 at telomeres in the mutant cells was observed (Supplementary Fig. 2e, f). In addition, chromatin immunoprecipitation (ChIP) assay confirmed the telomere enrichment of endogenous LSD1 in U2OS cells (Fig. 1k). More importantly, LSD1 enrichment at telomeres is reduced after siTERRA (Fig. 1k), confirming fluorescent imaging results (Fig. 1i, j). Taken together, these data are consistent with localization of LSD1 to ALT telomeres in a TERRA-dependent manner.

### LSD1 contributes to the ALT pathway independent of its catalytic activity
Given that LSD1 localizes to ALT telomeres, we next asked how LSD1 influences the ALT pathway. First, we investigated ALT features, including telomere clustering, APB number, and telomere DNA synthesis, after knocking down LSD1 with targeted siRNA verified by western blot (Fig. 2a). In contrast to control RNA, siLSD1 decreased the number and intensity of clustered telomeres both in G2 cells and TRF1-FokI cells, reflecting the impairment of telomere clustering (Fig. 2b, c, and Supplementary Fig. 3a, b). In addition, the APB number was also decreased in LSD1-silenced cells (Supplementary Fig. 3c, d). Furthermore, the number of FokI-induced Edu foci on telomeres decreased after siLSD1 (Fig. 2d, e), indicating impairment in telomere DNA synthesis. To confirm LSD1's role in ALT+ cells, we investigated the APB number and telomere clustering in other two ALT+ cell lines, Saos-2 and SK-N-FI, after siLSD1 (Supplementary Fig. 3e–h). Our results show that knocking down LSD1 leads to ALT defects in those cells as well. In all, these results suggest that the depletion of LSD1 decreases ALT activity.

Since LSD1 protects uncapped telomeres in telomerase-positive cells without demethylating histones[13], we sought to determine whether LSD1's contribution to ALT requires its catalytic activity. First, we treated TRF1-FokI cells with 200 nM pulrodemstat, a potent LSD1 demethylation inhibitor ($IC_{50} = 0.3$ nM, $EC_{50} = 7$ nM)[29]. We found that the inhibitor did not significantly affect DNA synthesis in FokI cells (Fig. 2f, g). We then assessed the density of H3K4me[2] and H3K9me[2], two LSD1 demethylating substrates[30,31], across ALT telomeres after siLSD1 with ChIP assay. We found H3K9me[2] and H3K4me[2] across telomeres were not significantly affected following siLSD1 with or without FokI-induced damage (Fig. 2h). Though H3K4me[2] was increased with FokI-induction for the 15q telomere, such increases were not affected

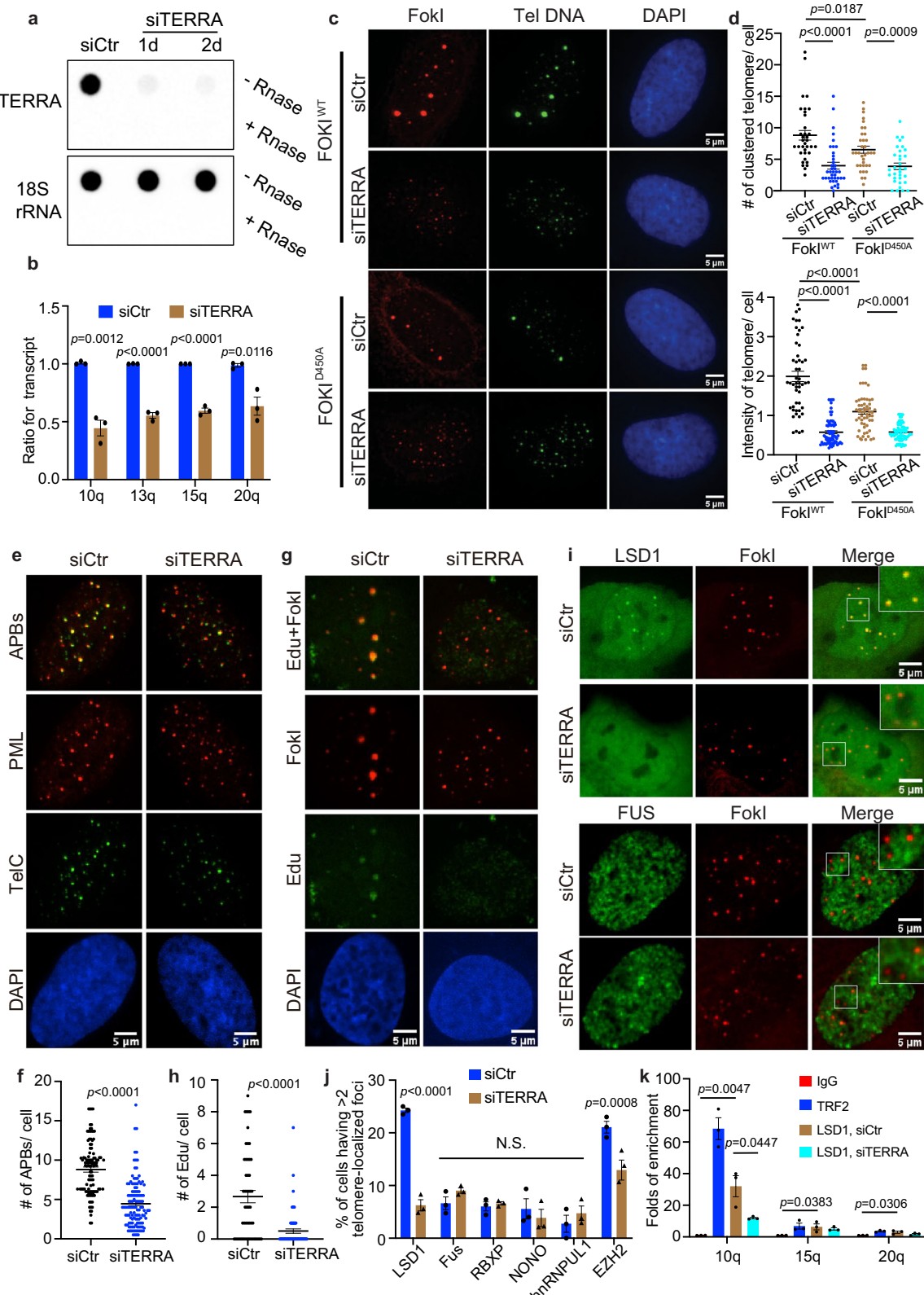

by siLSD1 (Fig. 2h, right panel). Taken together, data show that LSD1-catalyzed demethylation on nucleosomes is not an essential requirement for ALT telomere maintenance.

## LSD1 affects TERRA localization without altering its level

Next, we asked how LSD1 contributes to ALT beyond its catalytic activity. TERRA has been shown to act in trans by live imaging, wherein

TERRA is transcribed from a subset of telomeres and is recruited to other telomeres[7,32,33]. Since LSD1 interacts with TERRA in vitro and at uncapped telomeres[13,25], we examined whether LSD1 contributes to ALT through alteration of TERRA localization. After knocking down LSD1, we found decreased localization of TERRA on telomere in both G2 cells and TRF1-FokI cells (Fig. 2i, j, and Supplementary Fig. 3i, j). Given that LSD1 can potentially help to control TERRA transcription via

**Fig. 1 | LSD1 localization on ALT telomeres requires TERRA. a** Total TERRA levels were evaluated by northern-dot-blot assay, following transient knockdown of TERRA in U2OS for 24 and 48 h (*n* = two independent experiments). **b** TERRA levels from chromosome 10, 13, 15, 20 were quantitated by qRT-PCR, following transient knockdown of TERRA in U2OS for 48 h (*n* = three independent experiments, two-tailed unpaired *t* test). **c** Representative images and **d** quantification of telomere clustering from DNA FISH after siTERRA in U2OS cells stably expressing TRF1-FokI with treatment of 4-Hydroxyestradiol (4-OH, 6 h) (*n* = 50 cells for FokI^WT siCtr, *n* = 69 cells for FokI^WT siTERRA, *n* = 58 cells for FokI^D450A siCtr, *n* = 76 cells for FokI^D450A siTERRA over two independent experiments, two-tailed unpaired *t* test). **e** Representative images and **f** quantification of APB bodies (defined by PML and telomere FISH colocalization) after siTERRA in FokI cells (*n* = 110 cells for siCtr,

*n* = 122 cells for siTERRA over three independent experiments, two-tailed unpaired *t* test). **g** Representative images and **h** quantification of Edu foci on telomeres after siTERRA in FokI cells (*n* = 51 cells for siCtr, *n* = 70 cells for siTERRA over two independent experiments, two-tailed unpaired *t* test). **i** Representative images for GFP-LSD1 and GFP-FUS and **j** quantification of telomeric localization for GFP-tagged LSD1, FUS, NONO, RBXP, hnRNPUL1, and immunostained EZH2 with siControl and siTERRA in U2OS cells (more than 100 cells per group, *n* = three independent experiments, two-way ANOVA. **k** ChIP-qRCR assay was performed to detect telomeric LSD1 with or without siTERRA from indicated chromatin U2OS cells synchronized in G2 (*n* = three independent experiments, one-way ANOVA. Error bars are mean ± SEM. NS not significant. Source data are provided as a Source Data file.

its histone demethylase function, we tested TERRA levels through northern-dot-blot and qRT-PCR. The results showed that unlike siTERRA cells, which had reduced TERRA levels and therefore could serve as a positive control, the amount of TERRA did not significantly decrease in LSD1-depleted cells (Fig. 2k, and Supplementary Fig. 3k). These data indicate that LSD1 contributes to ALT via affecting TERRA localization and that LSD1 does not influence TERRA levels.

### LSD1 interacts with TERRA in U2OS cells via its nucleic acid-binding domain

So far, our data suggest that LSD1 and TERRA may exhibit mutual dependence for their localization on ALT telomeres. To test whether this requires TERRA-LSD1 interaction, we evaluated whether TERRA and LSD1 interact in U2OS cells. RNA immunoprecipitation assay (RIP) showed that TERRA was selectively immunoprecipitated by the LSD1 antibody and not the IgG control (Fig. 3a, Supplementary Fig. 4a). In addition, biotinylated TERRA pulled down exogenous LSD1 within U2OS lysates (Fig. 3b), supporting an TERRA-LSD1 interaction in U2OS cells.

Next, we tested which LSD1 domains contributed to its TERRA binding in U2OS cells. LSD1 consists of an intrinsically disordered region (IDR) at its N-terminal region (1–170 aa) followed by a SWIRM (Swi3p, Rsc8, and Moira) domain and an interleaved catalytic amine oxidase domain (AOD) domain (Fig. 3c)[30,34]. Biotinylated TERRA pulled down LSD1^ΔIDR (171–852 aa) which contains all structured domains[25,35], and was sligthly less effectively than the full length. On the other hand, TERRA could not pull down the IDR domain (Fig. 3b), suggesting that TERRA mainly binds LSD1's tertiary structure, consistent with biochemical, X-ray crystallography, and cross-linking mass spectroscopy data[13,25,36].

It has been shown that mutating LSD1 K355, K357, and K359 (termed LSD1^ΔIDR, 3KE), which forms a basic stripe along the amine oxidase domain, largely reduces the binding to extranucleosomal DNA[34]. This DNA binding region coincides with an extended RNA-binding site on the surface of LSD1[25]. The K661A mutation on AOD-C domain (hereafter named LSD1^ΔIDR, K661) is frequently used as a catalytically inactive mutant by disrupting the interaction with the FAD cofactor (Fig. 3c)[34]. We showed that TERRA pulled down LSD1^ΔIDR, K661 albeit not as strongly as the WT, suggesting that the calalytic domain conformation, like the IDR, might play a role in TERRA binding. Given that TERRA failed to pull down the RNA-binding mutant LSD1^ΔIDR, 3KE (Fig. 3b), we clonclude that LSD1's nucleic acid-binding region primarily interacts with TERRA.

### Recruitment of LSD1 is sufficient to enrich TERRA on ALT telomeres

Since TERRA localization is affected by many other proteins whose transcription could be regulated by LSD1, we sought to test whether LSD1's presence on telomeres is sufficient to enrich TERRA. Previously, we developed a small molecule-mediated protein heterodimerization system to recruit proteins to subcellular structures[37–40]. The dimerizer, TMP-Fluorobenzamide-Halo (TFH), consists of trimethoprim (TMP)

chemically linked to HaloTag ligand. TFH takes advantage of the selective interaction of TMP and HaloTag with the eDHFR and the HaloTag enzymes, respectively, to bring two fusion proteins into proximity (Fig. 3d). By fusing eDHFR to LSD1 and Halo to TRF1 and with the addition of the dimerizer TFH, we artificially recruited LSD1 to telomeres without activating the ALT pathway with DNA damage (Fig. 3e). Compared to the controls without adding the dimerizer (Supplementary Fig. 4b, c), recruitment of LSD1 to telomeres resulted in an increase of TERRA on ALT telomeres, indicating that the presence of LSD1 on telomeres is sufficient to enrich TERRA (Fig. 3e, f).

Next, we tested the LSD1 regions that are important for TERRA enrichment. Recruiting LSD1^ΔIDR, but not IDR domain alone, resulted in increased TERRA foci localization on telomeres (Fig. 3e, f, and Supplementary Fig. 4b, c). This suggests that the tertiary structure of LSD1^ΔIDR is sufficient to enrich TERRA on telomeres without DNA damage. In addition, recruiting the catalytically inactive mutant LSD1^ΔIDR, K661 can also enrich TERRA on telomeres, in agreement with a largely non-enzymatic role of LSD1 in ALT. By contrast, the nucleic acid-binding LSD1^ΔIDR, 3KE, exhibits a significantly lower percentage of TERRA localization at telomeres (Fig. 3e, f, and Supplementary Fig. 4b, c), indicating that DNA/RNA binding is required for LSD1-mediated TERRA enrichment on telomeres.

### LSD1 functional rescue requires LSD1 nucleic acid-binding, not demethylase activity

To confirm the importance of LSD1 domains in ALT function, we rescued ALT phenotypes in siLSD1-treated U2OS cells by artificially recruiting LSD1, its various truncations, the catalytically inactive mutant LSD1^ΔIDR,K661A, and its nucleic acid-binding mutant LSD1^ΔIDR,3KE to telomeres (Fig. 3c). Unlike the N-terminal IDR domain LSD1^IDR, the full-length LSD1^FL and C-terminal LSD1^ΔIDR both rescued the reduced telomere clustering in siLSD1 cells (Supplementary Fig. 4d, e). Similarly, telomere DNA synthesis in siLSD1 cells was restored after recruitment of LSD1^FL and LSD1^ΔIDR, but not the LSD1^IDR (Fig. 3g, h).

In contrast, the nucleic acid-binding mutant LSD1^ΔIDR, 3KE did not restore either telomere clustering or DNA synthesis in siLSD1 cells, whereas the catalytically inactive mutant LSD1^ΔIDR, K661A enhanced telomere clustering (Fig. 3g, h, and Supplementary Fig. 4d, e). Furthermore, dimerization of LSD1 to telomeres did not change the density of H3K4me^2, H3K9me^2, or H3K9me^3 on telomeres (Supplementary Fig. 4f), further suggesting that LSD1's demethylation activity on nucleosomes is not a requirement for initiation of the ALT pathway. Altogether, our results largely support a non-catalytic, organizational role of LSD1 in ALT through TERRA interactions.

### LSD1 phase separation on telomeres is enhanced by TERRA binding

Interestingly, we observed signatures of phase separation after recruiting LSD1 to telomeres. LSD1 foci fused together and continued to become bigger and brighter due to coarsening (Fig. 4a, Movie 1). Additionally, LSD1 foci fusion drove telomere clustering, resulting in fewer telomere foci (Fig. 4b). These data suggest LSD1

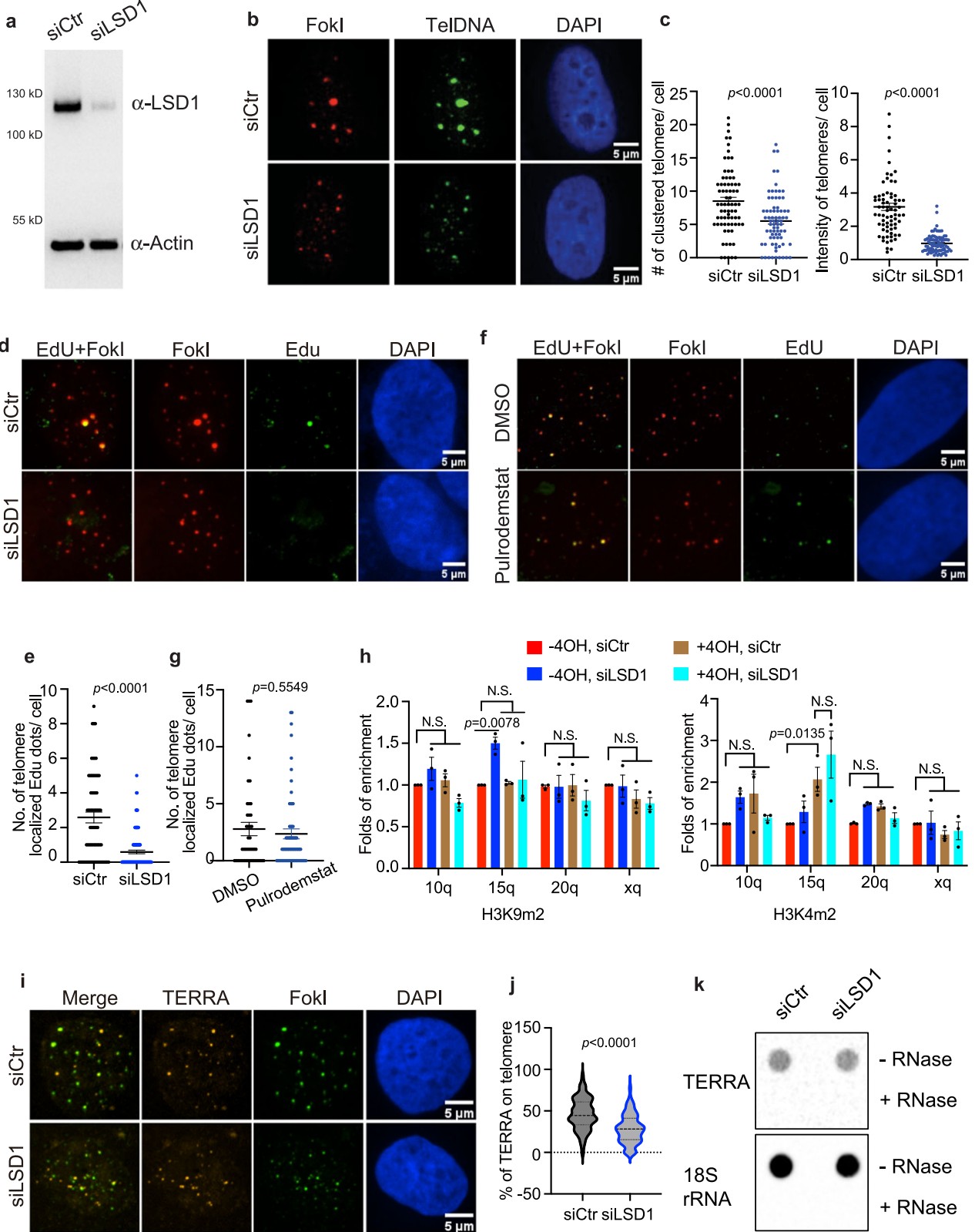

can undergo local condensation once concentrated on telomeres, similar to other proteins we have shown previously[39,40]. To test whether LSD1 phase separation at telomeres is ALT-specific, we recruited LSD1 to telomeres in ALT-negative HeLa cells. In contrast to U2OS cells, we did not observe signatures of phase separation such as LSD1 foci fusion or telomere clustering in HeLa cells (Supplementary Fig. 5a, Movie 2).

Since IDRs have been shown to mediate protein phase separation[41-43], we tested the importance of IDR domain (LSD1[IDR]) in LSD1 on ALT telomeres. However, recruiting LSD1[IDR] to telomeres in U2OS cells did not induce signature of phase separation and telomere clustering (Supplementary Fig. 5b, c, Movie 3). In contrast, the IDR deletion mutant LSD1[ΔIDR] was sufficient to drive phase separation and telomere clustering (Supplementary Fig. 5b, c, Movie 4). This indicates

**Fig. 2 | Knocking down LSD1 leads to loss of ALT phenotypes. a** Western blot of siLSD1 in U2OS cells (n = three independent experiments). **b** Representative images and **c** quantification of telomere clustering (DNA FISH) after siLSD1 in FokI stable cell line with treatment of 4-hydroxyestradiol (4-OH, 6 h) (n = 72 cells examined for siCtr, n = 90 cells for siLSD1 over two independent experiments, two-tailed unpaired t test). **d** Representative images and **e** quantification of Edu foci on telomeres after siLSD1 in FokI cells (n = 70 cells for siCtr, n = 102 cells examined for siLSD1 over three independent experiments, two-tailed unpaired t test). **f** Representative images and **g** quantification of Edu foci on telomeres after treatment of Pulrodemstat (200 nM, 16 h) in FokI cells (n = 46 cells for DMSO, n = 70 cells for Pulrodemstat over two independent experiments, two-tailed unpaired t

test). **h** ChIP-qRCR assay was performed to detect telomeric H3K4m² and H3K9m² from indicated chromatin in siCtr- or siLSD1-transfected FokI cells treated with and without 4-hydroxyestradiol (4-OH, 6 h) (n = three independent experiments, one-way ANOVA). **i** Representative images and **j** quantification of TERRA (RNA FISH) localization on telomeres after siLSD1 in FokI expressing U2OS cells (n = 136 cells for siCtr, n = 127 cells for siLSD1 over three independent experiments, two-tailed unpaired t test). **k** Northern-dot-blot was used to detect total TERRA levels in U2OS cells treated with siCtr or siLSD1 for 48 h (n = two independent experiments).18S rRNA serves as loading control. Error bars are mean ± SEM. NS not significant. Source data are provided as a Source Data file.

that a driving force for LSD1 phase separation arises from the structured core domain of LSD1 and that the IDR, by itself, is not sufficient for the observed phase separation properties in cells.

To test whether TERRA is important for LSD1 phase separation on telomeres, we knocked down TERRA and tracked telomere clustering after recruiting LSD1$^{\Delta IDR}$ to telomeres with TFH. LSD1$^{\Delta IDR}$ in siTERRA U2OS cells still showed signatures of phase separation and drove telomere clustering, but the degree of telomere clustering was much less compared to the control cells (Supplementary Fig. 5c, Movie 1 and 5). This indicates that the lack of TERRA does not abolish LSD1 phase separation but does substantially reduce the rate of LSD1 condensate fusion. This observation is consistent with previous work showing that LSD1 itself can phase separate in vitro[44]. Since LSD1 can interact with other G4 RNAs[13,25], it is also possible that LSD1 interacts with non-TERRA RNAs to undergo phase separation on telomeres.

Given that the DNA/RNA-binding mutant LSD1$^{\Delta IDR,\ 3KE}$ did not enrich TERRA on telomeres (Fig. 3e, f), we subsequently tested the ability of LSD1$^{\Delta IDR,\ 3KE}$ to phase separate in cells. Recruiting LSD1$^{\Delta IDR,\ 3KE}$ still showed signature of phase separation and fusion-driven telomere clustering (Fig. 4b, and Movie 6). However, recruiting LSD1$^{\Delta IDR,\ 3KE}$, much like siTERRA, induced less telomere clustering than recruiting LSD1$^{\Delta IDR}$ as shown through both time-lapse imaging analysis and DNA FISH quantification (Fig. 4b, and Supplementary Fig. 5d, Movie 6), suggesting that the TERRA-LSD1 interaction is indeed important for phase separation on telomeres. Based upon these results, we conclude that TERRA plays a role in LSD1 phase separation at ALT telomeres.

## TERRA promotes LSD1 phase separation in vitro

To more quantitatively assess how TERRA affects LSD1 phase separation, we examined TERRA and LSD1 phase behavior in vitro by purifying LSD1, in complex with CoREST for stability (Supplementary Fig. 6a, d) and synthesizing various repeats of 5'-UUAGGG-3' to mimic TERRA RNA (Supplementary Fig. 6b, c). Furthermore, with the H3K4me² nucleosome as substrates, we confirmed that LSD1 is enzymatically active (Supplementary Fig. 6e, f).

With 25 μM LSD1$^{FL}$ at 150 mM NaCl, the addition of TERRA resulted in the formation of spherical condensates (Fig. 4c). Mapping the phase diagram revealed that longer TERRA repeats can shift the LSD1 phase boundary to lower concentrations (Fig. 4d–f). In addition, reentrant behavior, where intermediate TERRA concentration drives LSD1 phase separation while low and high TERRA concentration inhibits LSD1 phase separation, was observed. This behavior has been observed in other RNA-driven phase separation examples[45,46]. Furthermore, a 20× (5'-UUAGGG-3') TERRA RNA can drive LSD1 to phase separate as low as 5 μM protein concentrations (Fig. 4f). Because endogenous TERRA can be long, with length estimated to be up to 9000 bases[47], we propose that long TERRA repeats will be highly effective at driving LSD1 phase separation at very low protein concentrations on telomeres.

We then tested the phase behavior for various LSD1 truncations with TERRA since our cell data showed that LSD1$^{\Delta IDR}$, but not the IDR, is sufficient for LSD1 phase separation. We purified LSD1$^{\Delta IDR}$ and confirmed it was enzymatically active with the H3K4me²

nucleosome as substrates (Supplementary Fig. 6d–g). Interestingly, LSD1$^{\Delta IDR}$ appears to be more efficient at phase separating with 8x TERRA than LSD1$^{FL}$ and even resulted in non-spherical condensates at low RNA and high protein concentrations (Fig. 4g, h and Supplementary Fig. 7a), suggesting a stronger interaction of TERRA with LSD1$^{\Delta IDR}$. An electrophoretic mobility shift assay (EMSA) indeed showed that a truncated LSD1$^{\Delta IDR}$ has stronger binding capability to TERRA ($K_d^{app}$ = 112 ± 7 nM) than full-length LSD1$^{FL}$ ($K_d^{app}$ = 1540 ± 90 nM) (Fig. 4i, and Supplementary Fig. 7b). This suggests that LSD1's IDR may play a regulatory role in LSD1-TERRA phase separation. Indeed, although IDRs in many RNA-binding proteins are shown to promote phase separation by providing interaction valence[48–51], IDRs in Ras GTPase-activating protein-binding protein 1 (G3BP1) are not sufficient to drive G3BP1 phase separation with RNAs for stress granule formation[52,53]. Instead, upon selective phosphorylation, IDRs in G3BP1 enable RNA binding of G3BP1 to promote phase separation. It is plausible that electrostatic mediated alterations such as phosphorylation, prevalent across the IDR of LSD1[54], may also influence the extent of protein-TERRA-mediated phase separation at ALT telomeres.

For the spherical LSD1-TERRA condensates, fusion events were observed (Fig. 4j), suggesting their liquid property. To further assess the material properties of TERRA-LSD1 condensates, we conjugated purified LSD1-CoREST with FITC and transcribed Cy3-Uridine labeled 12× TERRA. Fluorescence imaging showed uniform mixing of LSD1 and TERRA in the spherical condensates, lacking obvious substructures (Fig. 4k). FRAP (Fluorescence Recovery After Photobleaching) assay showed that the fluorescent intensity of TERRA recovered over time (Fig. 4l, m). The apparent diffusion coefficient, estimated from FRAP using $D_{app} = r^2/t$, where r is the radius of the photobleached region and t is the recovery time (Supplementary Table 1), is 0.01 ± 0.003 μm²/s. This value is comparable to other protein-RNA condensates[39]. This FRAP experiment also showed that the average mobile fraction is 58 ± 6%. This partial recovery indicates that LSD1-TERRA condensates are not simple liquids in which all components are mobile. The immobile fraction observed may be due to intermolecular interactions between TERRAs in the condensates. Since RNAs are known to aggregate once condensed and RNA helicases can prevent RNA aggregation in condensates[55], it is possible that RNA helicases are required to regulate TERRA-protein condensate fluidity at ALT telomeres.

Next, we tested the effect of nucleic acid-binding on LSD1 phase behavior. First, we found that the purified LSD1 nucleic acid-binding mutant LSD1$^{\Delta IDR,\ 3KE}$ abolished LSD1's ability to undergo phase separation with TERRA in vitro (Fig. 4n), which was consistent with the importance of RNA binding for LSD1 phase separation in cells (Fig. 4b). Since LSD1 affects TERRA localization but not TERRA level (Fig. 2i–k, Supplementary Fig. 3i, j), we also purified the LSD1 demethylation mutant LSD1$^{\Delta IDR,\ K661A}$ and examined its effect on LSD1 phase separation with TERRA. Unlike the nucleic acid-binding mutant LSD1$^{\Delta IDR,\ 3KE}$, the catalytically deficient mutant LSD1$^{\Delta IDR,\ K661A}$ maintains its ability to phase separate with TERRA (Fig. 4n).

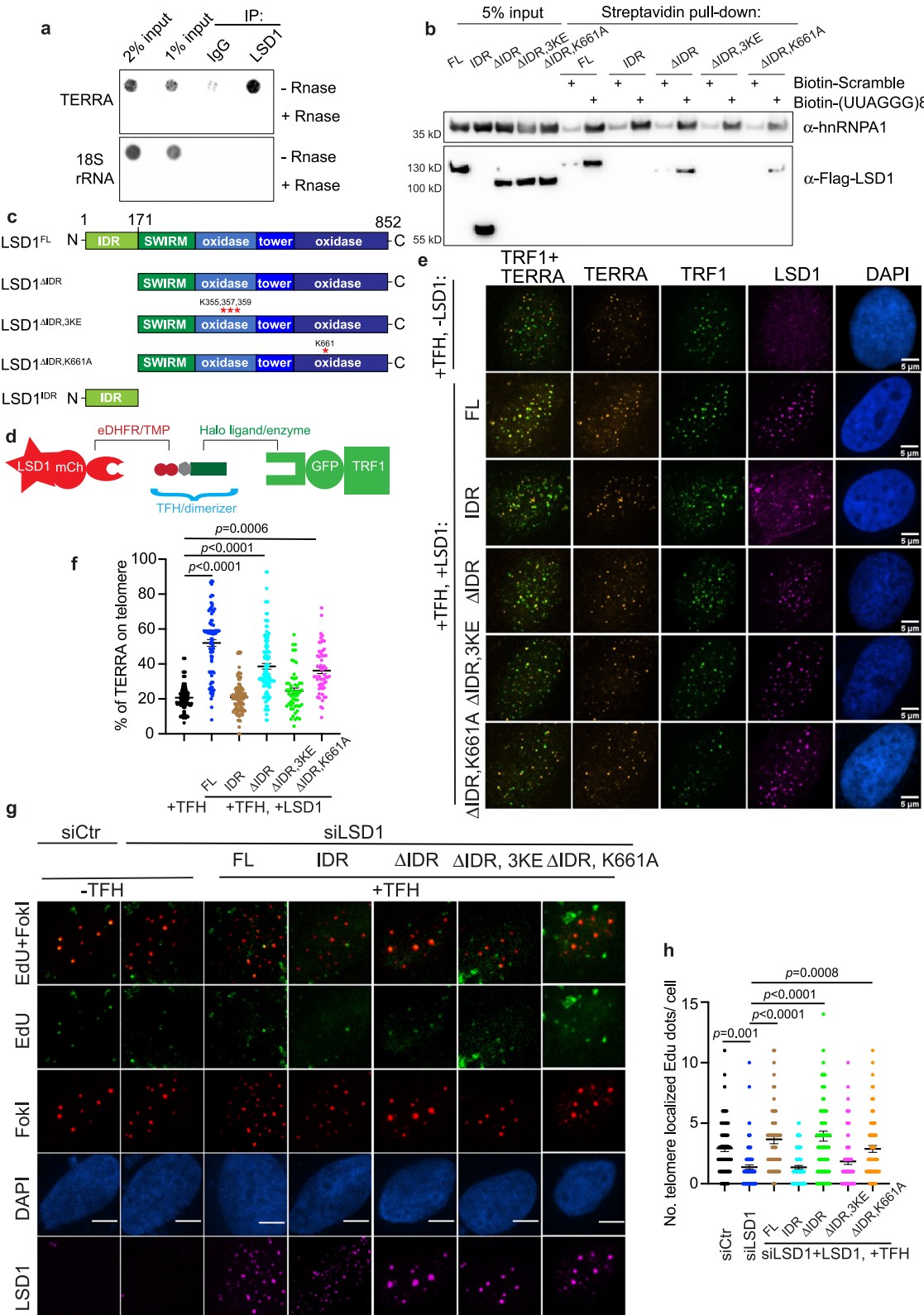

To confirm the binding between TERRA and these mutants, we used biotinylated TERRA to pull down the purified LSD1 mutants and observed that biotinylated TERRA can interact with both purified LSD1$^{\Delta IDR}$ and LSD1$^{\Delta IDR, K661A}$, but not with LSD1$^{\Delta IDR,3KE}$ (Fig. 4o). Overall, these data suggest that specific nucleic acid-binding properties of the protein are important for TERRA-stimulated LSD1 phase separation.

## TERRA G-quadruplex structure is important for LSD1 phase separation and ALT pathway

TERRA, with its guanine-abundant 5′-UUAGGG-3′ repeats, is prone to form extended parallel-stranded G-quadruplex structures[56–60]. Previous studies revealed that LSD1 preferentially binds to G-quadruplex TERRA[25], prompting us to ask whether TERRA-driven LSD1 phase separation is structure dependent. First, we used CD spectra to analyze

**Fig. 3 | Chemical dimerization of LSD1 to telomeres is sufficient to enrich TERRA and rescue ALT phenotypes. a** IP-recovered RNA was detected via dot-blot with probes annealing with TERRA or 18S rRNA (refer to Supplementary Fig. 4a for IP efficiency). **b** RNA pull-down assay was used to evaluate the binding of Streptavidin bead-bound biotinylated (UUAGGG)$_8$ oligonucleotide to flag-tagged LSD1$^{FL}$, LSD1$^{IDR}$, LSD1$^{\Delta IDR}$, LSD1$^{\Delta IDR, 3KE}$, and LSD1$^{\Delta IDR, K661A}$ from U2OS cell lysates (n = three independent experiments). **c** Schematic of LSD1: full-length (LSD1$^{FL}$), C-terminus (LSD1$^{\Delta IDR,}$), N-terminus (LSD1$^{IDR}$), mutant for nucleic acid binding (LSD1$^{\Delta IDR, 3KE}$), mutant for demethylation activity (LSD1$^{\Delta IDR, K661A}$). **d** Dimerization schematic: LSD1 is fused to mCherry and eDHFR, and TRF1 is fused to Halo enzyme and GFP. The dimerizer is TFH, TMP(trimethoprim)- Fluorobenzamide-Halo ligand. **e** Representative images and **f** quantification of TERRA (RNA FISH) localization on telomeres after dimerizing overexpressed LSD1$^{FL}$, LSD1$^{IDR}$, LSD1$^{\Delta IDR}$, and LSD1$^{\Delta IDR, 3KE}$, and LSD1$^{\Delta IDR, K661A}$ to Halo-TRF1 in U2OS cells with dimerizer TFH (n = 108 cells for +TFH alone, n = 87 cells for LSD1$^{FL}$, n = 103 cells for LSD1$^{IDR}$, n = 113 cells for LSD1$^{\Delta IDR}$, n = 55 cells for LSD1$^{\Delta IDR, 3KE}$, n = 53 cells for LSD1$^{\Delta IDR, K661A}$ over three independent experiments, one-way ANOVA). **g** Representative images and **h** quantification of Edu assay showing the rescue effect of newly synthesized telomeric DNA after dimerizing LSD1$^{FL}$, LSD1$^{IDR}$, LSD1$^{\Delta IDR}$, LSD1$^{\Delta IDR, 3KE}$, and LSD1$^{\Delta IDR, K661A}$ to telomeres in siLSD1 cells (n = 69 cells for siCtr, n = 89 cells for siLSD1, n = 67 cells for LSD1$^{FL}$, n = 77 cells for LSD1$^{IDR}$, n = 90 cells for LSD1$^{\Delta IDR}$, n = 75 cells for LSD1$^{\Delta IDR, 3KE}$, n = 81 cells for LSD1$^{\Delta IDR, K661A}$ over two independent experiments, one-way ANOVA). Error bars are mean ± SEM. NS not significant. Source data are provided as a Source Data file.

the folding of the synthesized TERRAs and found all RNAs to assemble into G-quadruplex structures (Supplementary Fig. 8a, b). We then compared LSD1 phase separation with TERRA and other non-G-quadruplex forming RNAs. Three single-strand RNAs (anti-FAM57, anti-Myo1b, and PolyU), which have a similar length to 8x TERRA but are predicted to contain no known structure, served as controls. In addition, the large non-coding RNase P RNA represented a control for non-G-quadruplex forming RNA. RNase P is G-rich and contains an elaborate tertiary structure[61]. Both microscopic imaging and turbidity measurement of absorbance at 340 nm showed that LSD1 phase separation is only produced by TERRA, but not by other non-G-quadruplex containing RNAs (Fig. 5a, b). To further confirm that LSD1 phase separation is G-quadruplex dependent, we tested LSD1 phase separation with other two G-quadruplex RNAs, LMNB and HIRA[62]. Both of the G-quadruplex RNAs could drive LSD1 condensation, with HIRA formed spherical condensates and LMNB formed irregular shaped aggregates, perhaps owing to the differences in the properties of their G-quadruplex that affect LSD1 binding (Supplementary Fig. 9a, b). Furthermore, to control for loss of TERRA G-quadruplex, we changed the GGG sequence in the 8x TERRA to GAG and used a 3x UUAGGG RNA that would disrupt G-quadruplex formation. Unlike the 8x TERRA, those two negative control RNAs did not drive LSD1 phase separaton (Supplementary Fig. 9a, b). All together, our data indicates that TERRA-mediated LSD1 phase separation is specific to G-quadruplex RNA conformations.

To further assess TERRA structure-dependent phase separation, we used N-methyl mesoporphyrin IX (NMM), a small molecule that binds parallel-stranded G-quadruplexes[63], to treat TERRA before testing phase separation with LSD1. NMM treatment with a low dosage facilitated TERRA-LSD1 condensate transition into a more solid phase, as indicated by the morphological change from spherical to irregular shape (Fig. 5c). Moreover, a high dosage of NMM fully aggregated TERRA-LSD1 condensates (Fig. 5c). To undertand this result, we first confirmed NMM binding to TERRA using the shift in the UV absorbance spectrum of NMM after adding TERRA (Supplementary Fig. 8c). Next, we used EMSA to evaluate how NMM impacts the LSD1-TERRA interaction. The LSD-TERRA complex caused a marked mobility shift (Fig. 5d) and it appeared that NMM treatment led to a more compact gel shift of the TERRA-LSD1 complex (Fig. 5d). This indicates that NMM may stabilize the TERRA-LSD1 interaction to enhance LSD1 phase transition from droplets to an aggregated state (Fig. 5c). The putative stabilizing effect of NMM on TERRA G-quadruplex formation is similar to other chemical probes that are known to target G-quadruplex RNAs[64,65,66].

Based on this observation, we next treated cells with NMM to see how manipulating TERRA G-quadruplex affects the ALT pathway. The FokI-induced cells treated by NMM had fewer clustered telomeres than control cells (Fig. 5e, f), suggesting the impairment of ALT machinery, possibly due to disrupted fluid condensates. Since NMM treatment has been shown to slightly decrease TERRA expression in cells[63] (Supplementary Fig. 8d), we used an alternative approach to investigate the role of TERRA's structure in the ALT pathway. This involved dimerizing

the RNA helicase DHX36, which specifically recognizes and dissolves RNA G-quadruplex[67,68] at telomeres. We reasoned that dimerizing DHX36 to telomeres would allow DHX36 to locally dissolve TERRA G-quadruplex in a specific manner. With this method, we indeed observed reduced telomere clustering (Fig. 5g, h). Further, transfection of PolyC, which was reported to suppress G-quadruplex RNA folding[68], also inhibited telomere clustering (Fig. 5i, j). Due to difficulties in observing TERRA G-quadruplex formation inside of dynamic U2OS cells, it is a challenge to directly assess the specificity and selectivity of any of the treatments on G-quadruplex formation. However, the collective results from the three treatments all support the notion that TERRA's tertiary fold plays an important role in the ALT pathway, likely through modulation of RNA structure-based phase separation with LSD1.

## LSD1 condensates promote R-loop formation on telomeres
Since TERRA contributes to ALT telomere maintenance through R-loop formation[8,21] and we showed that LSD1 enriches TERRA on telomeres (Fig. 3e, f), we asked whether LSD1 plays a role in R-loop formation. To visualize R-loops in living cells, we fused GFP with tandem repeats of HBD, the N-terminal hybrid-binding domain of RNase H1, which specifically recognizes DNA–RNA hybrids in a non-sequence-specific manner[69,70]. With this method, R-loop foci visualized by GFP fused 2x HBD showed an increase on telomeres after dimerizing LSD1$^{\Delta IDR}$ to telomeres (Fig. 6a, b). To verify that this method can selectively detect R-loops, we further co-expressed RNaseH as a negative control. RNaseH overexpression, which is known to dissolve R-loops[71], largely decreased GFP-HBD foci on telomeres, indicating that the signal probed by GFP-2x HBD represents an R-loop conformation. In addition, the nucleic acid-binding mutant LSD1$^{\Delta IDR, 3KE}$ was less efficient in forming R-loops (Fig. 6a, b), presumably due to the inability of LSD1$^{\Delta IDR, 3KE}$ to enrich TERRA on telomeres (Fig. 3e, f). These data further support that LSD1 promotes R-loop formation in a manner that involves LSD1-TERRA binding.

To confirm the effect of LSD1 on R-loop formation, we employed the TeloDRIP assay to quantify the amount of R-loops on telomeres in a population of cells[72]. As a positive control, elevated R-loop levels on telomeres, especially from chromosome 10 and 13, were observed after FokI-trigged telomere damage (Fig. 6c). However, knocking down LSD1 largely suppressed the production of R-loops by FokI, indicating that LSD1 is important for R-loop formation. As a negative control, RNaseH treatment abolished R-loop signals (Fig. 6c). Importantly, depletion of TERRA significantly decreased R-loop levels, confirming that these R-loops contain TERRA. In accordance with HBD R-loop markers in living cells, the TeloDRIP assay also revealed that dimerizing LSD1$^{\Delta IDR}$ increased R-loops on telomeres (Fig. 6d). By contrast, dimerizing the nucleic acid-binding mutant LSD1$^{\Delta IDR, 3KE}$ was less efficient in promoting R-loop formation, further suggesting the direct involvement of TERRA in LSD1-mediated R-loop formation (Fig. 6d). Taken together, our data support a model whereby LSD1 promotes R-loop formation via enriching TERRA on telomeres.

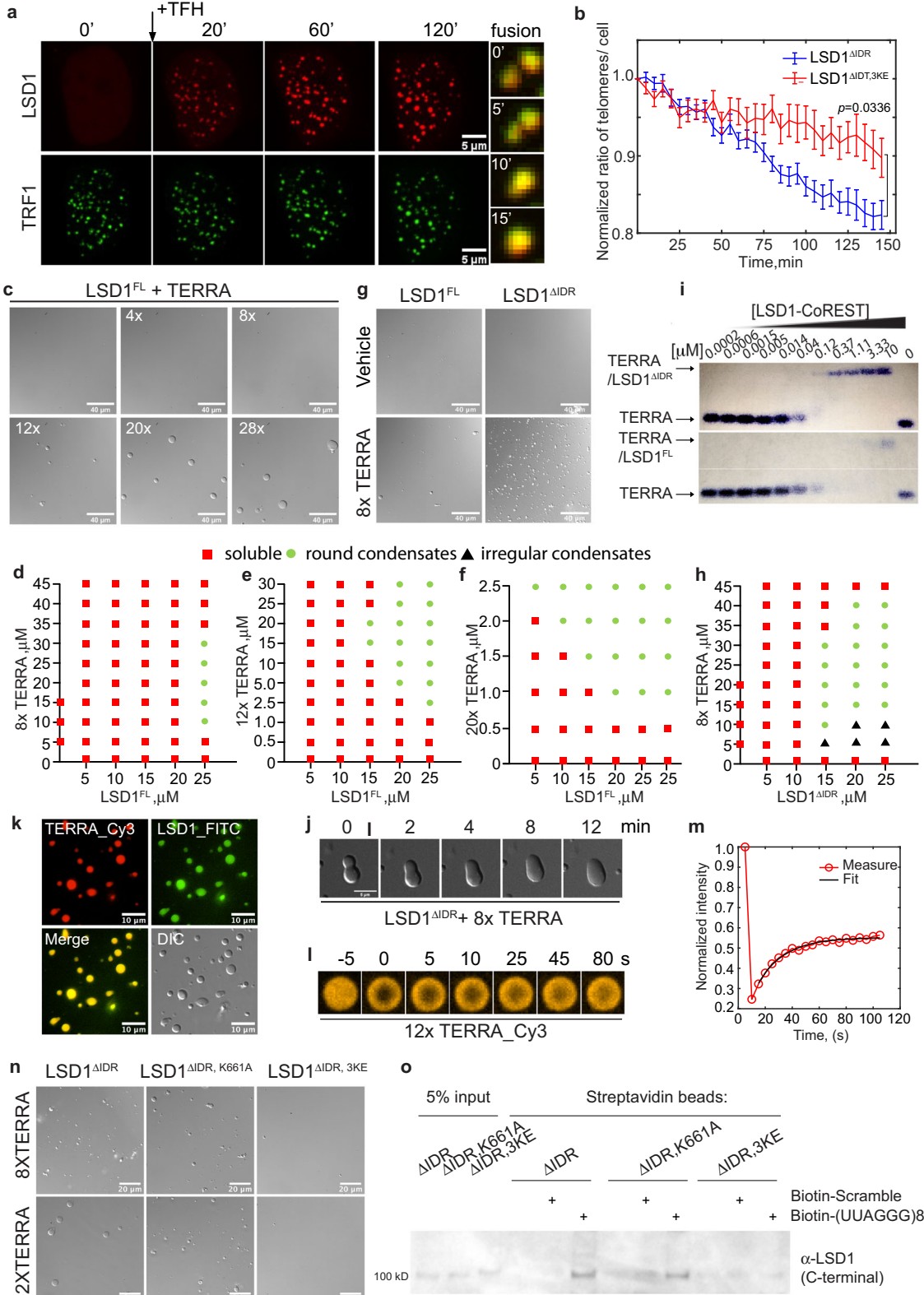

## LSD1-TERRA condensates enrich R-loop stimulating protein Rad51AP1

Previous work showed that DNA repair protein Rad51 and RAD51-associated protein Rad51AP1 help TERRA R-loop formation by invading dsDNA[22,24]. We suspected that in addition to enriching TERRA, LSD1-TERRA condensates could potentially enrich R-loop stimulating proteins on telomeres to promote R-loop formation. To test this possibility, we examined whether LSD1-TERRA condensates can enrich

Rad51AP1 in vitro and in cells. First, we combined purified Rad51AP1 (Supplementary Fig. 10a) with LSD1 and TERRA separately and discovered that Rad51AP1 did not phase separate with LSD1, but it did make irregularly shaped condensates with TERRA (Fig. 6e). Next, we combined Rad51AP1 with TERRA and LSD1 together, which resulted in condensates containing both Rad51AP1 and LSD1. Interestingly, these condensates maintained the round shape and were present in greater quantities than those formed by LSD1-TERRA or TERRA-Rad51AP1

**Fig. 4 | TERRA drives LSD1 phase separation. a** Representative images of U2OS cells expressing mCh-eDHFR-LSD1 and Halo-GFP-TRF1 after adding TFH to induce dimerization at indicated time points. Montage on the right shows the fusion event of LSD1 foci over time, every 5 min between images. **b** Mean telomere number per cell after dimerizing mCh-eDHFR-LSD1 (WT and 3KE mutant) to Halo-GFP-TRF1 stably expressed in U2OS cells ($n = 20$ cells for LSD1$^{\Delta IDR}$, $n = 18$ cells for LSD1$^{\Delta IDR, 3KE}$ over three independent experiments, two-tailed unpaired $t$ test). **c** Representative DIC images of condensates formed by a mixture of purified LSD1 at 25 μM and different UUAGGG repeats as mimics of TERRA at 2.5 μM (n = three independent experiments). d-f Phase diagram of full-length LSD1 with the addition of 8× (**d**), 12× (**e**), 20× (**f**), TERRA. **g** Representative DIC images and **h** phase diagram of LSD1$^{\Delta IDR}$ with 8× TERRA. **i** Representative EMSA showing (UUAGGG)$_8$U RNA binding of truncated LSD1 CoREST (top) and full-length LSD1-CoREST (bottom) (n= two independent experiments). **j** Fusion of condensates containing 25 μM LSD1 and 25 μM 8× TERRA. **k** Fluorescent images of condensates containing Cy3-labeled 10 μM 12× TERRA and FITC-labeled 50 μM LSD1 (n = three independent experiments). **l** FRAP of condensates containing 10 μM 12× Cy3-TERRA and 50 μM LSD1 condensates. **m** A representative FRAP curve for TERRA is shown. The fit to an exponential curve generates mobile fraction and recovery time. Mean values of 19 condensates from three independent experiments are shown in Supplementary Table 1. **n** Representative DIC images of LSD1$^{\Delta IDR}$, LSD1$^{\Delta IDR, 3KE}$, and LSD1$^{\Delta IDR, K661A}$ with the addition of TERRA (n = three independent experiments). **o** In vitro pull-down assay was used to evaluate the binding of biotinylated (UUAGGG)$_8$ oligonucleotide to recombinant LSD1$^{\Delta IDR}$, LSD1$^{\Delta IDR, 3KE}$, and LSD1$^{\Delta IDR, K661A}$ from *E. coli*. (n = three independent experiments). Source data are provided as a Source Data file.

alone. This suggested that Rad51AP1 and LSD1 have a synergistic effect in phase separating with TERRA, perhaps by interacting through an extended protein-RNA-protein network.

To recapitulate the conditions in cells, we dimerized LSD1 to telomeres to form LSD1 condensates in the presence of TERRA and observed that a small fraction of cells showed enrichment of Rad51AP1 on telomeres (Supplementary Fig. 10b, c). This modest enrichment may be due to the lack of DNA damage regulated Rad51AP1 post-translational modifications, or the lack of direct Rad51AP1 recruitment to telomeres through other recombination proteins. To confirm the dependence of Rad51AP1 telomeric localization on LSD1 and TERRA, we performed a ChIP assay to compare the enrichment of Rad51AP1 on telomere after treating cells with siLSD1 and siTERRA. We observed a decrease in Rad51AP1 enrichment in both cases (Fig. 6f).

To confirm that Rad51AP1 contribute to LSD1-stimulated R-loop formation, we measured R-loop in Rad51AP1 knockout cells by DRIP after enrichment of LSD1[73]. Compared with Rad51AP1 WT cells, the elevated R-loop levels by LSD1 enrichment is dramatically decreased in Rad51AP1 knockout cells (Fig. 6g), indicating that Rad51AP1 contributes to LSD1's function in promoting R-loop formation in ALT cells. However, Rad51AP1 deficiency did not abolish LSD1 induced R-loop formation, suggesting other proteins might also be enriched in the LSD1-TERRA condensates to facilitate R-loop formation. Nevertheless, our data support that Rad51AP1 enriched in LSD1-TERRA condensates can aid R-loop formation

### LSD1-TERRA interaction affects ALT cell viability
So far our data suggest that LSD1-TERRA phase separation facilitates R-loop formation for ALT telomere maintenance. To further determine the long term effect of LSD1-TERRA interaction in ALT cells, we monitored the cell viability of LSD1 knocking down ALT cells. SiLSD1 impaired cell growth and colony formation in U2OS cells (Fig. 7a–c). However, overexpression of LSD1$^{\Delta IDR}$, but not LSD1$^{\Delta IDR, 3KE}$ could rescue the defects of cell survival (Fig. 7b, c). Together, these results support the notion that LSD1-TERRA interaction is critical for ALT cell viability.

## Discussion
A well-established role of TERRA in ALT pathway is to form R-loops to promote DNA damage and or form RD-loops to facilitate DNA synthesis[8,9,22,24]. Here we show that TERRA contributes to ALT through interacting with LSD1. Our work suggests TERRA can recruit LSD1 to telomeres in response to DNA damage-based ALT signaling. The LSD1-TERRA interaction network on telomeres, likely enhanced by DNA damage-mediated LSD1 or TERRA modifications, can trigger local LSD1-TERRA condensate formation (Fig. 7d). This in turn may help to enrich TERRA and other TERRA interacting proteins, such as R-loop stimulating protein Rad51AP1, to promote R-loop formation.

The precedence for TERRA-containing R-loops in ALT telomere maintenance has been established, where an excess of TERRA and R-loop formation can result in telomere loss[74]. Therefore, to promote productive telomere DNA synthesis in ALT, TERRA and R-loop levels in ALT must be tightly regulated, likely through multiple pathways. On one hand, TERRA levels are shown to be fine-tuned via conventional RNA polymerase II-governed transcription as well as the RNase-mediated RNA decay pathway[6,8,75]. On the other hand, a production/degradation-independent pathway via recombinase protein stimulation of R-loop formation has been reported[9,22,24]. Our work provides another production/degradation-independent pathway to spatially regulate R-loop formation by enriching TERRA and R-loop stimulating proteins via phase separation. In demonstrating that the TERRA RNA tertiary structure is important for TERRA phase separation and function on ALT telomeres, we speculate that RNA helicases that stabilize or dissolve G-quadruplex may act as R-loop regulators.

LSD1 is a multifaceted protein that functions to remove methyl groups from monomethyl or dimethyl lysine residues, such as histone 3 lysine 4 (H3K4me1/2) or non-histone proteins such as p53[30,54,76–78]. However, LSD1 also has demethylase-independent activities such as mediating protein stability and scaffolding chromatin modifiers or transcription factors[77,79]. Recent work shows that LSD1 also phase separates with epigenetic readers and transcription factors (ZMYND8, BRD4, and FOXA1) at super-enhancers to regulate chromatin function[44,80]. Here we show an additional enzymatic activity-independent function of LSD1 that uses LSD1's ability to interact with G-quadruplex RNA TERRA to promote R-loop formation in ALT cells. It is worth mentioning that LSD1 enzymatic activity has been linked to R-loop inhibition formation at the transcriptionally targeted region by demethylating H3K4 to help to facilitate recruitment of helicase DDX19A[81]. Thus, LSD1 could help to dissolve R-loops to enable DNA repair in ALT cells, although the dynamic or transient nature of LSD1 activity may make it hard to detect. Previous work showing that G-quadruplex RNA binding inhibits LSD1 demethylation activity[25] suggests that LSD1 could play a non-catalytic role in R-loop formation via its association with TERRA and R-loop formation. Recently, it has been shown that multi-domain chromatin remodeling complexes within the polycomb group (PcG), such as PRC1 and the GQ RNA associated PRC2 complex, can also undergo phase separation to potentially enhance chromatin compaction[82,83]. Thus, it seems that methyl regulatory histone complexes that interact with RNA structure may utilize similar phase separation-based strategies to facilitate chromatin organization.

Considering the compact interaction network at telomeres, it is not surprising that chromatin-associated RNAs can regulate functions via binding to chromatin modification enzyme complexes[84,85]. The increasing evidence that RNA might phase separate with chromatin-associated proteins to create local compartments on chromatin provides interesting perspectives on gene function, telomere organization, and dynamic RNP assemblies[45,86,87]. Repetitive RNAs, such as TERRA, are well equipped to fulfill a myriad of biological roles at chromatin because of the ability to hybridize with

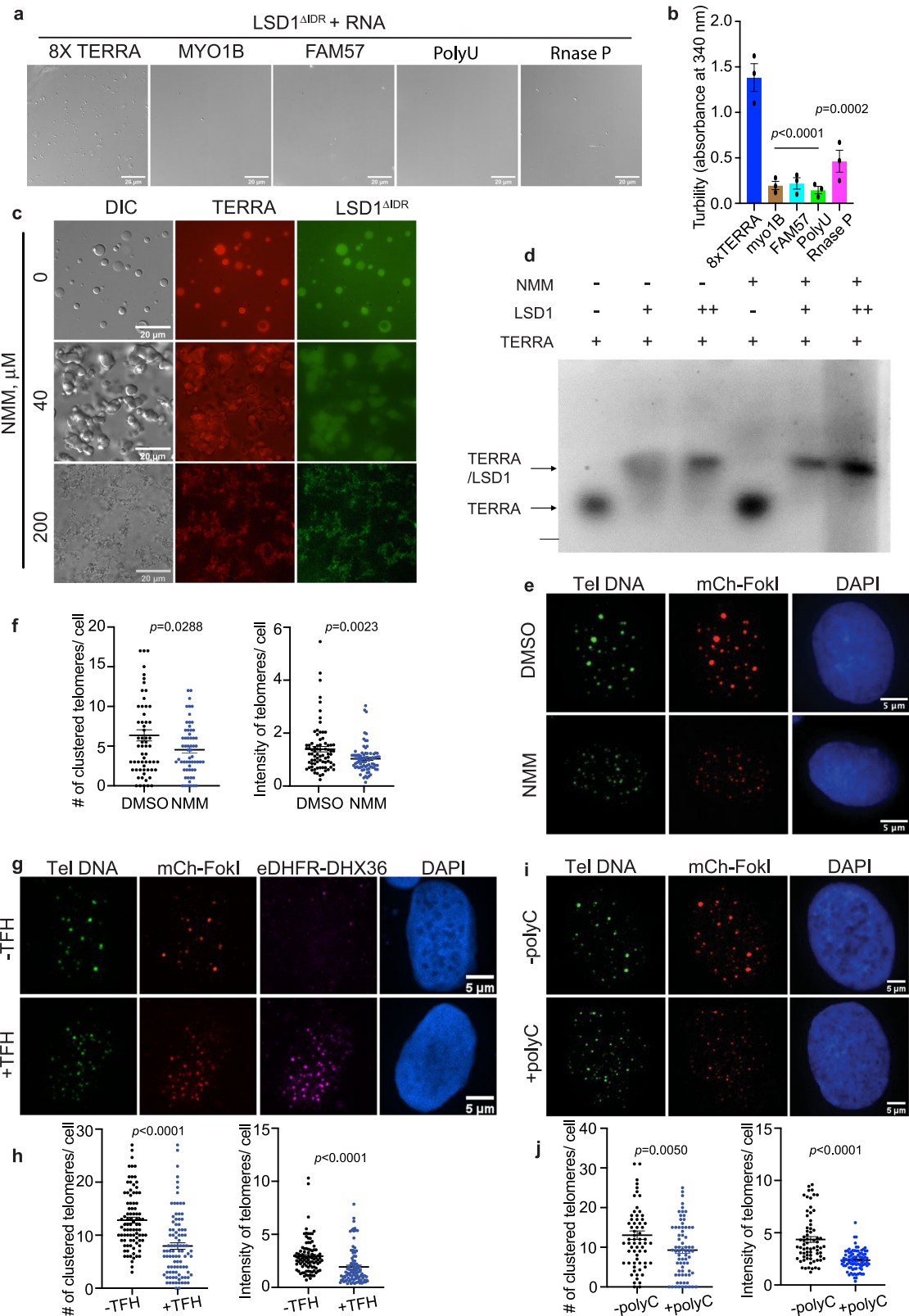

local DNA in a sequence-specific manner and assemble into a higher order RNA quaternary structure. These roles may impart efficient nucleation control that allows for specificity in condensate formation[88–91]. It is likely that other repetitive RNA elements that contain structures may also interact with chromatin modifiers to promote local phase separation, facilitating the compartmentalization on chromatin to regulate genome stability.

## Methods

### Cell culture

Majority of experiments were performed with U2OS cells, except otherwise indicated with Saos-2, SK-N-FI and Hela1.3. U2OS cells stably expressing 3x Halo-GFP-TRF1 were created previously in the lab[39], and mCherry-TRF1-FokI (WT, D450A) were gifted by Roger A. Greenberg[28]. Cells were cultured in growth medium (Dulbecco's Modified Eagle's

**Fig. 5 | TERRA G-quadruplex structure is required for LSD1 phase separation and ALT functions.** Representative DIC images (**a**) and turbidity assay (**b**) of LSD1$^{\Delta IDR}$ (20 μM) with the addition of different RNAs (20 μM). Turbidity measurements are shown as mean ± SEM (n = three independent experiments, one-way ANOVA). **c** Representative images of fluorescent LSD1$^{\Delta IDR}$ (50 μM) with 12× TERRA (25 μM) pretreated with G-quadruplex binding ligand, N-methyl mesoporphyrin IX, (NMM) (n= three independent experiments). **d** In vitro EMSA of 8X TERRA GG(UUAGGG)$_8$U (+ = 2 μM) alone and in the presence of LSD1 (+ = 10 μM and ++ = 20 μM) and NMM (+ = 200 μM). The 0.75% agarose gel contained Syber Safe dye (10,000:1, Invitrogen) and was performed under the following conditions (60 V, 1 h, 4 °C). The fluorescence data were visualized at 520 nm using an Amersham Imager 600 (AI600) instrument (n = two independent experiments). **e** Representative images and **f** quantification of telomere clustering (DNA FISH) after treatment of NMM (200 nM, 24 h) in FokI stable cell line (n = 77 cells for DMSO, n = 80 cells examined for NMM over two independent experiments, two-tailed unpaired t test). **g** Representative images and **h** quantification of telomere clustering (DNA FISH) after co-transfection of eDHFR-DHX36, Halo-TRF1 and mCh-FokI with or without TFH treatment in U2OS cells (n = 96 cells for -TFH, n = 99 cells examined for +TFH over three independent experiments, two-tailed unpaired t test). DHX36 are indicated by antibody-based immunostaining. **i** Representative images and **j** quantification of telomere clustering (DNA FISH) after co-transfection of polyC with mCh-FokI in U2OS cells (n = 69 cells for -polyC, n = 84 cells examined for +polyC over three independent experiments, two-tailed unpaired t test). Error bars are mean ± SEM. N.S., not significant. Source data are provided as a Source Data file.

medium with 10% FBS and 1% penicillin–streptomycin) at 37 °C in a humidified atmosphere with 5% $CO_2$. For G2 cell synchronization, cells were first treated with 2 mM thymidine for 21 h, released into fresh medium for 4 h, and then further treated with 15 μM CDK1 inhibitor RO3066 for 12 h.

## Plasmids
The plasmid for inducing DNA damage at telomeres (mCherry-TRF1-FokI) was previously published[28]. To construct the plasmids 2x Flag-eDHFR-LSD1(1-852 aa, 1-170 aa,171–852 aa), GFP-eDHFR-LSD1(1-852 aa, 1-170 aa,171–852 aa), and mCherry-eDHFR-flag-LSD1(WT), the full length and fragmental LSD1 were amplified from Addgene plasmid #109157 and introduced into target plasmids through in-fusion cloning (Takara Bio). For the plasmids of 2x Flag-eDHFR-LSD1 (K661A, K355/357/359) and mCherry-eDHFR-flag-LSD1 (K661A, K355/357/359), mutational LSD1 were amplified from the original plasmids gifted by Dr. Song Tan[34]. The other plasmids, including GFP-tagged NONO, FUS, hnRNPUL1, RBXP were gifted by Dr. Roderick J. O'Sullivan. For the plasmids of GFP-2x HBD, 2x HBD were amplified from the original plasmids gifted by Dr. Kaiwei Liang and subcloned into the target backbone. All the target plasmids in this study are derived from a plasmid that contains a CAG promoter for constitutive expression, obtained from E. V. Makeyev[92].

## siRNA transfection
All the siRNAs were purchased commercially, and the sequences are listed in Supplementary table 2. Two siRNAs targeting LSD1 3'UTR as a pool were transfected at 50 nM with Lipofectamine 2000 (Invitrogen) 48 hr prior to imaging, following the manufacturer's protocol. TERRA knockdown by ASO was done as described[7]. Briefly, cells were transfected with 100 nM of scrambled or anti-TERRA ASO and RNAiMax (ThermoFisher) diluted in OptiMEM (Life Technologies). Transfection medium was replaced with complete culture media 12 h later, and cells were imaged at 24 h post transfection.

## TERRA fluorescence and telomere DNA fluorescence in situ hybridization (FISH)
TERRA FISH assay was performed as previously described[38]. Briefly, cells were washed twice with cold PBS and treated with cytobuffer (100 mM NaCl, 300 mM sucrose, 3 mM $MgCl_2$, 10 mM PIPES pH 7, 0.1% Triton X-100, and 10 mM vanadyl ribonucleoside complex) for 7 min on ice. Cells were fixed in 4% formaldehyde for 10 min at room temperature, followed by permeabilization and blocking in PBS buffer containing 0.5% Triton X-100, 1% BSA and 10 mM vanadyl ribonucleoside complex for 30 min. Cells were then dehydrated in a series of ethanol washes 70, 85, and 100% for 5 min each at room temperature, and the coverslips were dried at room temperature. 20 nM Telo Miniprobe Cy3 short probe provided by Dr. Bruce Armitage (Carnegie Mellon University, PA, USA) in hybridization buffer (50% formamide, 2× SSC, 2 mg/ml BSA, 10% dextran sulfate) was added to coverslips and then placed in a humidified chamber at 39 °C overnight. The following day, the coverslips were washed in 2× SSC with 50% formamide three times at 39 °C for 5 min each, three times in 2× SSC at 39 °C for 5 min each, and finally one time in 2× SSC at room temperature for 10 min. The coverslips were mounted on glass microscope slides with Vectashield mounting medium containing DAPI and analyzed with microscopy.

For the combined immunofluorescence DNA FISH, cells were rinsed with PBS and treated with cytobuffer for 2 min on ice, followed by fixation in 4% paraformaldehyde for 10 min at room temperature. Cells were then permeabilized and blocked in PBS buffer containing 0.5% Triton X-100, 1% BSA for 30 min at room temperature. Cells were then incubated with the indicated antibodies diluted in PBS buffer for 2 hr at room temperature. Following incubation with primary antibody, the cells were washed three times with PBST (PBS containing 0.1% Triton) for 10 min each and subsequently incubated with secondary antibody for 1 hr at room temperature. The cells were washed three times with PBST for 10 min each and then fixed in 4% paraformaldehyde for 10 min at room temperature. Cells were then dehydrated in a series of ethanol washes 70%, 85%, 100% for 2 min each at room temperature, and the coverslips were dried for 10 min. 100 nM PNA-TelC-Alexa 488 probe (F1004, PANAGENE) in hybridization buffer (70% formamide, 10 mM Tris 7.4, 0.5% Roche blocking solution) was added to coverslips and DNA was denatured at 75 °C for 3 min and then place in humidified chamber at room temperature overnight. The following day, the coverslips were washed with washing buffer (70% formamide, 10 mM Tris 7.4) three times for 5 min each, three times in PBST for 5 min each. The coverslips were mounted on glass microscope slides with Vectashield mounting medium containing DAPI and analyzed using fluorescence microscopy. All the primary antibodies we used for immunofluorescence staining are purchased from commercial company, including anti-TRF1 (Fisher, PIPA5111273), anti-flag (Cell Signaling, #14793), anti-mCherry (Fisher, PIMA532977), anti-EZH2 (Cell Signaling, #5246), and anti-PML (Santa Cruz, sc966).

## Protein purification and fluorescent labeling
Human full-length (1-852 aa) or ΔIDR (171–852 aa) LSD1 with CoREST (286–482 aa) protein were co-expressed in Rosetta (DE3) pLysS competent cells using either pET-15b (FL-LSD1) or pGEX-6P-1-ΔIDR LSD1 plasmids with pET28-CoREST plasmid, as previously described[25] with the following modifications. Rosetta (DE3) pLysS competent cells were grown at 30 °C in auto induction media containing 200 μg/mL ampicillin, kanamycin, and 40 μg/mL chloramphenicol antibiotic concentrations. After sonication, cell lysate was purified using Ni-affinity chromatography and GST affinity chromatography. After restriction-grade thrombin and 3C precision protease digestion to remove histidine and GST tags, an additional GST affinity chromatography column and a subsequent Superdex 200 size exclusion chromatography were used to further purify the LSD1-CoREST complex. In addition, two different point mutant constructs of LSD1 containing K661A or K355E/K357E/K359E with CoREST (286–482 aa), pST44STRaHISNhLSDD1x32-hCORESTD2 and pST44STRaHISNhLSDD1x17-hCORESTD2, respectively, were purified as described[34] and were a gift from Dr. Song Tan. In all cases, protein was stored in 25 mM HEPES Na pH 7.5, 100 mM

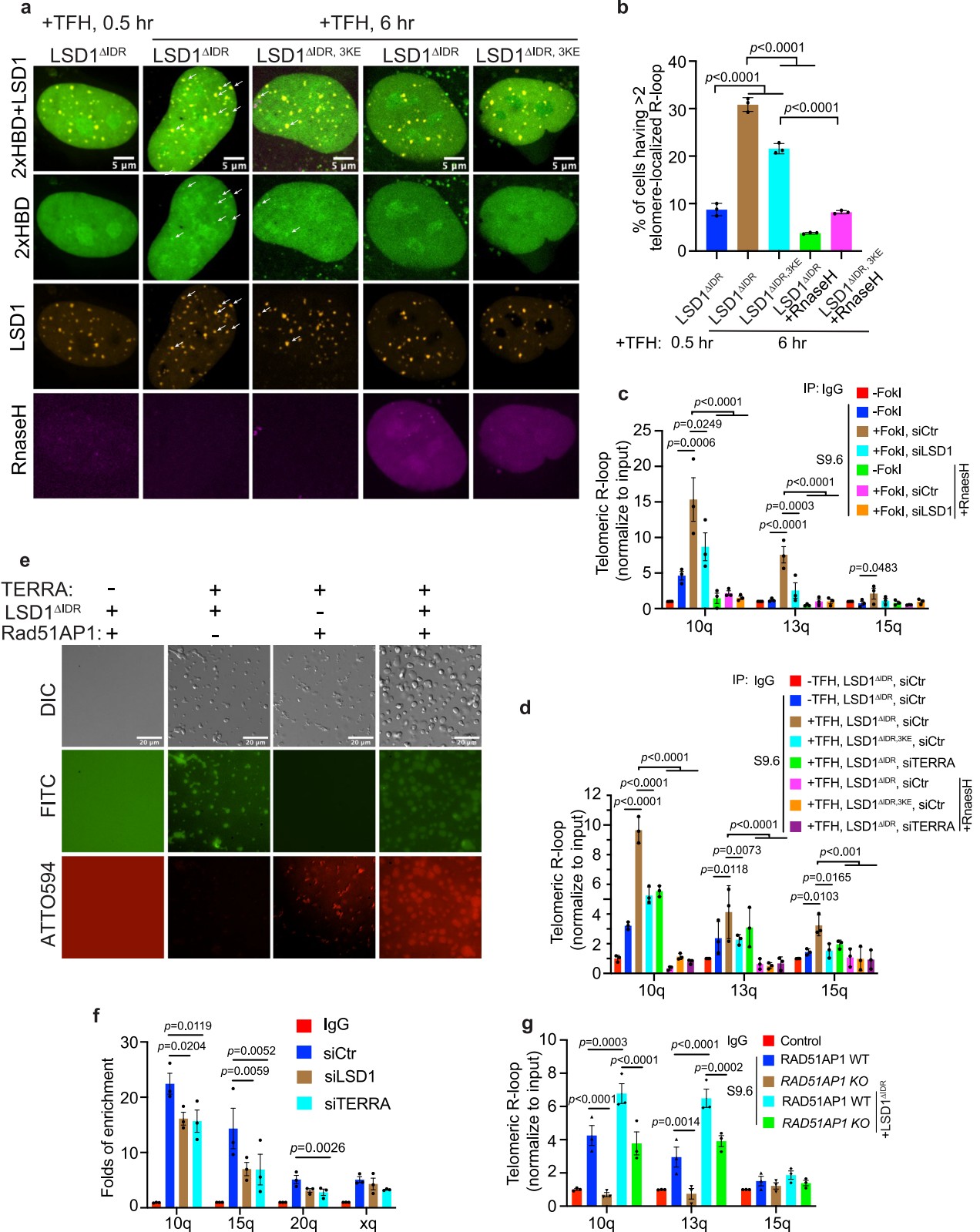

NaCl, and 5 mM TCEP buffer. The concentration of protein samples was determined by the BCA method (BCA Protein Assay kit, Pierce) with bovine serum albumin as a standard and validated with UV-VIS-spectroscopy, using the extinction coefficient for the FAD cofactor of LSD1 at 450 nm ($\varepsilon = 11300\,M^{-1}\,cm^{-1}$). Protein spectra were recorded on a Shimadzu UV-2600 Spectrophotometer and protein purity was assessed by SDS electrophoresis with Coomassie staining.

To express MBP and His-fused Rad51AP1, His tag was fused to the C-terminus end of Rad51AP1 and cloned into the pMal.C2X vector. For Rad51AP1 purification, MBP-Rad51AP1-His proteins expressed in bacterial BL21 (DE3) cells induced by 0.3 mM IPTG (Isopropyl β-D-1-thio-galactopyranoside) for 48 h at 16 °C and collected by sedimentation. The *Escherichia coli* cells were suspended in binding buffer (50 mM Tris-Cl pH 7.4, 500 mM NaCl and 10 mM Imidazole) and lysed with

**Fig. 6 | LSD1 promotes R-loop formation at ALT telomeres. a** Representative images and **b** quantification of R-loop (indicated by 2× HBD-GFP) on telomeres after dimerization of LSD1$^{ΔIDR}$, LSD1$^{ΔIDR, 3KE}$ to Halo-TRF1 in U2OS cells (n = three independent experiments, one-way ANOVA). Because the telomeres are not labeled, LSD1 foci after adding TFH for 0.5 h (shortly after dimerization completion) were used to infer telomere locations. **c** DRIP assay shows R-loop level on telomeres from chromatin 10q, 13q, 15q after siLSD1 or siTERRA (100 nM, 48 h) in FokI stable cell line (n = three experiments, one-way ANOVA). **d** DRIP assay shows R-loop level on telomeres from chromatin 10q, 13q, 15q after dimerization of LSD1$^{ΔIDR}$ and LSD1$^{ΔIDR, 3KE}$ to Halo-TRF1 in siCtr or siTERRA U2OS cells (n = three independent

experiments, two-way ANOVA). **e** Fluorescent images of condensates after combination of ATTO594-labeled Rad51AP1 (30 μM), FITC-labeled LSD1$^{ΔIDR}$ (50 μM) and 8× TERRA (50 μM). **f** ChIP-qRCR assay detecting Rad51AP1 binding to telomere DNA from indicated chromatin in siCtr-, siLSD1-, and siTERRA- transfected U2OS cells (n= three independent experiments, one-way ANOVA). **g** DRIP assay shows R-loop level on telomeres from chromatin 10q, 13q, 15q after dimerization of LSD1$^{ΔIDR}$ to Halo-TRF1 in U2OS RAD51AP1 WT and *KO* cells (n = three independent experiments, two-way ANOVA). Error bars are mean ± SEM. NS not significant. Source data are provided as a Source Data file.

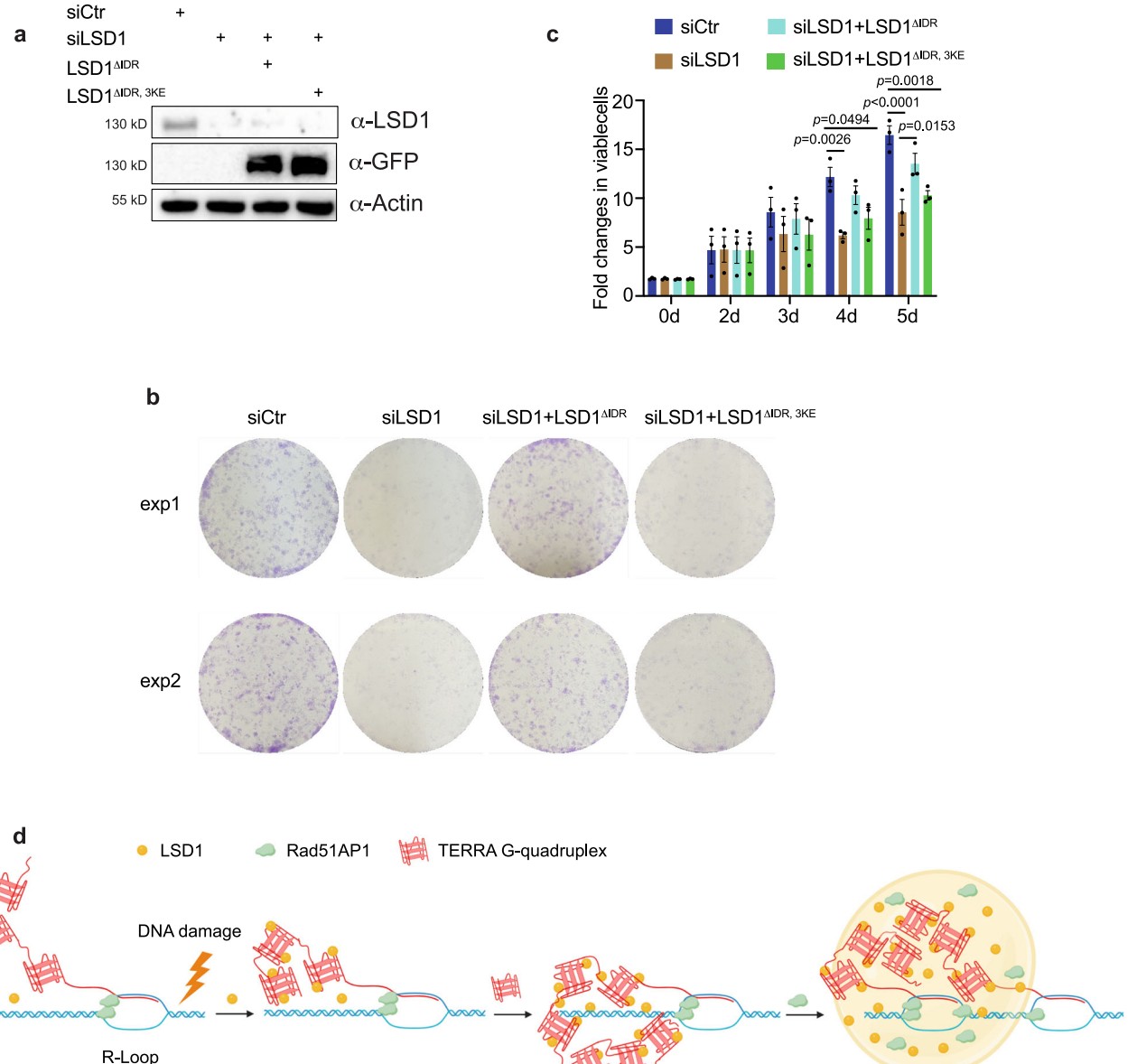

**Fig. 7 | LSD1-TERRA interaction affects ALT cell viability. a** Western blot of GFP-tagged LSD1$^{ΔIDR}$ and LSD1$^{ΔIDR,3KE}$ expression in siLSD1 cells (n = three independent experiments). **b** Images of colony formation assays with U2OS cell after indicated treatment. **c** Quantification of viable U2OS cells after indicated treatment using CellTiter-Blue assay (n = three independent experiments, one-way ANOV. Error bars are mean ± SEM). **d** A model for TERRA-LSD1 phase separation at ALT telomeres. In response to DNA damage signaling in the ALT pathway, TERRA recruits LSD1 to

telomeres. TERRA-LSD1 interaction at telomeres, likely enhanced by DNA damage, may further enrich TERRA on telomeres, leading to local condensate formation. Hereafter, other proteins, such as Rad51AP1, may partition into TERRA-LSD1 condensates to help stimulate R-loop formation for telomere maintenance in ALT cancer cells. Created with BioRender.com. Source data are provided as a Source Data file.

multiple cycles of sonication (Branson SFX250 Sonifier) and sedimented at 12,000 rpm for 30 min to pellet the debris. The supernatant lysates were purified using Ni-NTA Sepharose beads (GE Healthcare). After extensive washing with binding buffer, factor Xa protease (New England Biolabs) was incubated with recombinant protein bound to sepharose beads with a concentration of 50 µg ml$^{-1}$ beads for 4 h at room temperature. After extensive washing, the Rad51AP-His protein was eluted from beads in elution buffer (50 mM Tris-Cl pH 7.4, 500 mM NaCl, and 500 mM Imidazole). The Rad51AP1-His protein was further purified through the dialysis cassette (Thermo Fisher, 66380) to remove imidazole. Finally, the purified proteins were concentrated by centrifugal filtrations (Milllipore), then stored in aliquots at −80 °C.

LSD1 was labeled with fluorescence using FITC labeling kit (Millipore, 343210), following the manufacturer's protocol. Rad51AP1 was labeled with fluorescence using Atto 594 protein labeling kit (Sigma-Aldrich, 68616), following the manufacturer's protocol.

## TERRA RNA synthesis

Various TERRA RNA containing 4×, 8×, 12×, 20×, or 28× UUAGGG RNA repeats, as well as ssRNAs, including anti-FAM57B RNA (28 mer), anti-MYO1B RNA (33 mer), Poly U RNA (32 mer), were purchased as DNAs from IDT (Integrated DNA Technologies, see Supplementary Table 2), transcribed, PAGE-UREA purified, and desalted using Bio Rad desalting spin columns. RNAs were diluted in 10 mM Tris−HCl pH 7.4, 1 mM TCEP and 100 mM KCl and folded using a standard protocol (2 min at 95 °C, 5 min at 85 °C, 5 min at 75 °C, 5 min at 55 °C, 15 min at 37 °C, and then placed on ice). For RNA duplex formation, RNAs were annealed by mixing equimolar amounts of RNA oligonucleotide solutions in nuclease free water were mixed in annealing buffer (10 mM Tris−HCl pH 7.4, 50 mM NaCl 5 mM MgCl$_2$). The solution was heated at 94 °C, 4 min and gradually cooled down to room temperature for duplex formation. Previous folding protocols were used to anneal the bacterial RNase P RNA from *Thermotoga maritima*[93–95], which contains a high degree of tertiary structure and serves as a control RNA. In addition, The Cy3-labeled 12X TERRA RNA was purified and prepared in-house as described above, with an altered in vitro transcription reaction that contained UTP (3 mM) and the Cy3-labeled UTP (0.8 mM) (APEx-BIO). Global conformations of each RNA were evaluated using circular dichroism (CD) spectroscopy, where RNAs were recorded at room temperature on a Chirascan™ V100 CD spectrophotometer with a 1 mm cell, 1 nm band width, 4 s per point, and 0.1 ms timed intervals.

## In vitro phase separation assay

The recombinant protein was added to solutions at varying concentrations with indicated final salt in buffer (25 mM HEPES pH 7.5, 100 mM NaCl, 2 mM TCEP). The TERRA RNA was added into protein solution at indicated concentration and mixed thoroughly. The protein and RNA solution was then immediately loaded onto a homemade chamber comprising silicone isolator (VWR, 100490-928) and a glass slide (Fisher Scientific 22-266-858p) with a coverslip. Slides were then imaged with a Nikon confocal microscope with a 100x objective. Unless indicated, images represent condensates that have settled on the glass coverslip. Turbidity analysis was done using Nalo Drop by measuring the absorbance at 340 nm.

## FRAP assay

The fluorescently labeled LSD1 and TERRA RNA were added at indicated concentration and mixed thoroughly. The protein and RNA solution was then immediately loaded onto the glass slide treated with 1% BSA with a coverslip. Slides were then imaged with a Nikon confocal microscope using 100x objective images and were visualized with the 561 nm laser channel at 5 second intervals up to 85 s pre- and post exposure to droplet bleaching with 561 nm laser stimulation at 40%

power for 40 ms exposure duration. Images were processed and analyzed using NIS-Elements Software to obtain mobile fractions, recovery time, and diffusion coefficients.

## Quantitative real-time PCR

Total RNAs were isolated using RNA miniprep (Zymo, NC1047980), and 1 µg RNAs were used for reverse transcription using iScript cDNA Synthesis kit (Cat# 1708891, Bio-Rad). Quantitative PCR reaction were done with equal amounts of cDNA using a DyNAmo SYBR Green qPCR kit (Cat# F410, ThermoFisher Scientific). Signals were detected by QuantStudio 3 (Thermo Fisher A28567). GAPDH was used as the internal control. The sequences of individual primers for the selected telomeres are listed in Supplementary Table 2.

## Cell viability assays

For the colony formation assay, 2000 U2OS cells were seeded in 6 well plates in duplicate and cultured for 10 days before fixation and staining in a 1% crystal violet solution. Plates were images to show the positive stained colonies per well. To dynamically measure cell culture viability, cells were measured in 96-well microplate format using CellTiter-Blue® cell viability assay kit (Promega, G8081), following the manufacturer's protocol. Fluorescent signals were measured by a CLARIOstar microplate reader (BMG LABTECH).

## DNA−RNA immunoprecipitation (DRIP)

DRIP was performed according to the protocol[72]. For LSD1 knock down assays, siRNAs were transfected in mCherry-TRF1-FokI cells. Cells were induced with doxycycline (40 ng/mL$^{-1}$) 24 h, followed by tamoxifen (1 mM) and Shield (1 mM) 6 h before harvest. Approximately $5 \times 10^6$ cells per group were harvested by trypsinization and placed on ice. Cells were washed with 1× PBS and centrifuged (500 × g, 5 min, 4 °C). Cells were then dissolved in 200 mL cold RLN buffer (50 mM Tris−HCl pH 8.0, 140 mM NaCl, 1.5 mM MgCl$_2$, 0.5% NP-40, 1 mM dithiothreitol (DTT), and 100 U/mL RNasin Plus (Promega N2615) on ice for 5 min. Cells were centrifuged (500 × g, 5 min, 4 °C), and the liquid was carefully discarded. After washing the nucleic containing pellet in 500 µl cold PBS, cell pellets were lysed in 500 mL RA1 lysis buffer (NucleoSpin RNA, Macherey-Nagel) plus 5 µL of β-mercaptoethanol and homogenized through a needle (0.9 mm × 40 mm). The homogenized extracts were mixed with 250 mL H$_2$O and 750 mL ultra-pure phenol: chloroform: isoamyl alcohol (25:24:1). The samples were transferred to a heavy phase-lock tube and centrifuged (13,000 × g, 5 min). The top aqueous phase was poured into a tube containing 50 mM NaCl and 750 mL ice-cold propanol. The samples were shaken vigorously followed by incubating on ice for 30 min until nucleic acids precipitated. The samples were centrifuged (10,000 x g, 30 min, 4 °C) to obtain nucleic acid pellets. The pellets were washed twice with 70% ice-cold EtOH and air-dried. The pellets were dissolved in 135 mL of H$_2$O and sonicated with microtips for 200 and 400 Watt sonifiers (Branson Ultrasonics™ Sonifier™ SFX250) to obtain fragments of approximately 200 bp. Thirty micrograms of nucleic acids was incubated with RNaseH or H$_2$O in RNaseH buffer (Fisher Scientific, FEREN0201) at 37 °C for 90 min. Protein G beads (Sigma-Aldrich, GE17-0618-01) were washed three times in DIP1 Buffer (10 mM HEPES-KOH pH 7.5, 275 mM NaCl, 0.1% Na-deoxycholate, 0.1% SDS, 1% Triton X-100). Samples were diluted by ten in DIP1 Buffer, and 40 µL of ProteinG beads were added for 1 hour of pre-clearing rotating at 4 °C. One percent of the sample was saved as input. Samples were transferred to a new tube, and 40 µL of Protein G beads with 3 µg of S9.6 antibody were added to each sample and incubated overnight, rotating at 4 °C. Next, beads were washed with DIP2 Buffer (50 mM HEPES-KOH pH 7.5, 140 mM NaCl, 1 mM EDTA pH 8.0, 1% Triton X-100, 0.1% Na-deoxycholate), DIP3 Buffer (50 mM HEPES-KOH pH 7.5, 500 mM NaCl, 1 mM EDTA pH 8.0, 1% Triton-X 100, 0.1%Na-deoxycholate), and DIP4 Buffer (10 mM Tris−HCl pH 8.0, 1 mM EDTA pH 8.0, 250 mM LiCl, 1% NP-40, 1%

Na-deoxycholate). The sample was eluted in 100 mL of Elution Buffer (50 mM Tris−HCl pH 8.0, 10 mM EDTA pH 8.0, 0.5% SDS, Proteinase K) shaking at 55 °C for 2 h. DNA was purified (Qiagen, 28104) and eluted in 100 μL H$_2$O.

### RNA immunoprecipitation (RIP)

RIP assay was performed as described[24]. Briefly, U2OS cells (approximately $5 \times 10^7$ per condition) were harvested by trypsinization, washed, and dissolved in 500 mL of cold RLN buffer (50 mM Tris−HCl pH 8.0, 140 mM NaCl, 1.5 mM MgCl$_2$, 0.5% NP-40, 1 mM DTT, and 100 U/ml RNasIN PLUS). After centrifugation ($13,000 \times g$, 10 min, 4 °C), the inputs (10%) for each sample were collected from the supernatant. Samples were then incubated with 5 μg of antibodies overnight, followed by washing and incubation with Protein A/G beads for another 4 hr rotating at 4 °C. The beads were washed with RLN wash buffer for 5 min each. After 4 times wash, samples were eluted with 100 mL of elution buffer (5 mM EDTA, 1% SDS, and 10% β-mercaptoethanol) at 42 °C shaking at 1500 RPM for 30 min and at 65 °C for additional 30 min. RNA from the elution was isolated using RNA miniprep (Zymo, NC1047980), and half of each sample was digested with RNase for at least 3 h. After denatured in denature buffer (50% Formamide, 15% Formaldehyde, and 20 mM MOPS Buffer) at 65 °C for 15 min, the inputs and IP samples were then dot-blotted onto a positively charged Nylon membrane (Sigma, 11209299001) for detection. Samples were cross-linked to the membrane (UV Stratalinker 2400, Stratagene). The membrane was prehybridized in a hybridization buffer (Ultrahyb Ultrasensitive Hybridization Buffer, Invitrogen) rotating at 42 °C for 1 h. The membrane was hybridized with Biotin-labeled (TAACCC)$_7$ or Biotin-labeled (CGGAACTACGACGGTATCTG) 18S oligonucleotides at 42 °C overnight. The next day, the membrane was detected using Chemiluminescent Nucleic Acid Detection Module Kit (Thermo 89880), according to the manufacturer' instructions.

### Chromatin immunoprecipitation assay (ChIP)

The ChIP assay was conducted as described using a SimpleChIP Enzymatic Chromatin IP Kit[96]. Briefly, ~$4 \times 10^6$ cells were fixed with 1% formaldehyde for 10 min and quenched in glycine for additional 5 min at room temperature. Cells were lysed in extraction buffer on ice to obtain nuclear pellet, followed by incubation with micrococcal nuclease (1 μl, 20 min at 37 °C) to fragment genomic DNAs. To ensure complete lysis of the nucleus, further sonication was carried out with 3 sets of 20-s pulses. The resulting sheared DNAs were subjected to immunoprecipitation using normal IgG, anti-Histone H3 (dimethyl K9) Abs (ab1220), anti-Histone H3 (dimethyl K4) Abs (Millipore, 07-030), anti-Histone H3 (tri-methyl K9) (D4W1U), or anti-Rad51AP1 Abs (proteintech, 11255-1-AP). Quantitative real-time PCR analysis was performed to assess the immunoprecipitated telomere DNAs. The sequences of oligos used for ChIP-qPCR are listed in Supplementary Table 2.

### RNA pull-down assay

To bind labeled RNA to beads, 100 pmol of biotinylated TERRA or scamble RNA diluted in RNA capture buffer (20 mM Tris pH 7.5, 1 M NaCl, 1 mM EDTA) were incubated with 50 μL of streptavidin magnetic beads for 30 min at room temperature with agitation. Before incubation with RIP extract, beads were blocked with 1% BSA for 1 hr. After 3 times wash, beads were incubated with 200 ug RIP extract from U2OS cells expressing WT LSD1 and mutant LSD1 for 2 h at 4 °C in protein-RNA-binding buffer (20 mM Tris pH 7.5, 50 mM NaCl, 2 mM MgCl$_2$, 0.1% Tween™−20). Beads were washed 5 times and samples were subjected to SDS-PAGE and Western blotting.

### Telomere DNA synthesis detection by EdU

Following transfection, cells were pulsed with EdU (10 μM) for 1 hr before harvest. Cells on glass coverslips were washed twice in PBS and fixed with 4% paraformaldehyde (PFA) for 10 min. Cells were permeabilized with 0.3% (v/v) Triton X-100 for 5 min. The Click-IT Plus EdU Cell Proliferation Kit with Alexa Flour 488 (Invitrogen) was used to detect EdU.

### Protein dimerization with chemical dimerizers

The dimerizer TFH (**T**MP- **F**luorobenzamide-**H**alo) is used and its synthesis and storage was reported previously[37]. Dimerization on telomeres was performed as previously described[38]. Briefly, TFH was added directly to growth medium to a final working concentration of 100 nM. Cells were incubated with the dimerizer-containing medium for the indicated times, followed by immunofluorescence (IF) or fluorescence in situ hybridization (FISH). For movies with protein dimerization, the dimerizers are first diluted to 200 nM in growth medium and then further added to cell chambers to the working concentration after first-round imaging.

### Cell imaging

Imaging acquisition was performed as previously described[38]. For live imaging, cells were seeded on 22 × 22 mm glass coverslips coated with poly-D-lysine (Sigma-Aldrich). When ready for imaging, coverslips were mounted in magnetic chambers (Chamlide CM-S22-1, LCI) with cells maintained in normal medium supplemented with 10% FBS and 1% penicillin/streptomycin at 37 °C on a heated stage in an environmental chamber (TOKAI HIT Co., Ltd). Images were acquired with a microscope (ECLIPSE Ti2) with a 100 × 1.4 NA objective, a 16 XY Piezo-Z stage (Nikon Instruments Inc.), a spinning disk (Yokogawa), an electron multiplier charge-coupled device camera (IXON-L-897), and a laser merge module that was equipped with 488, 561, 594, and 630 nm lasers controlled by NIS-Elements Advanced Research. For both of fixed cells and live imaging, images were taken with 0.5 μm spacing between Z slices, for a total of 8 μm. For movies, images were taken at 5 min intervals for up to 3 h.

### Image processing

Images were processed and analyzed using NIS-Elements Software. Maximum projections were created from z stacks, and thresholds were applied to the resulting 2D images to segment and identify TERRA foci as binaries. For colocalization quantification of two fluorescent labels, fixed images were analyzed by the binary operation in NIS-Elements AR. Colocalized foci were counted if the objects from different layers containing overlapping pixels.

The degree of telomere clustering is analyzed with a custom MATLAB (Mathworks) code to define a size cutoff for telomere clusters from DNA FISH. This is to account for heterogeneous sizes for ALT telomeres so large telomere foci would not automatically be counted as telomere clusters. In each experiment setting, diameters of telomere foci were fitted into a two-component Gaussian mixture model using the MATLAB fitgmdist function. The mean of the larger component was calculated for the experiment and control group and the smaller value was use as the cutoff value. Foci in each group larger than the cutoff value were defined as telomere clusters.

### Electrophoretic mobility shift assay (EMSA)

To initiate binding reactions, the RNA (0.25 μM), such as the GG(UUAGGG) 8 U (8× TERRA) RNA, was incubated in the presence of increasing amounts of the LSD1-CoREST complex (0, 0.25, 1.25, 2.5, 5, 12.5, 25 μM). In addition, EMSA studies of 8X TERRA (0.25 μM) and LSD1-CoREST (+ = 10 μM and ++ = 20 μM) were incubated in an identical manner with and without NMM (+ = 200 μM). After incubation at 4 °C for 5 mins, reactions were loaded immediately into a 0.75% agarose gel containing 1110 Syber Safe dye (10,000:1 dilution), 1× THE (pH 7.4, 34 mM Tris-base, 66 mM HEPES, 0.1 mM EDTA), and 0.1 M potassium acetate. Gels were run for 60 min at 6 V/cm in 1× THE running buffer supplemented with 10 mM potassium acetate, with

constant buffer recirculation at 4 °C. RNA-binding studies to either the LSD1$^{\Delta IDR}$-CoREST, the catalytically inactive mutant (K661A, A539E) LSD1$^{\Delta IDR}$-CoREST, or the nucleosome binding site mutant (K355E, K357E, K359E) LSD1$^{\Delta IDR}$-CoREST complexes were performed using (0, 0.25, 1.25, 2.5, 5, 12.5, 25 μM) protein concentrations. Emission maxima fluorescence imaging of the gel as detected at 520 nm (Amersham Imager 600 (AI600)), enabling the binding reaction profile to be quantified, and only signal corresponding to fully bound or unbound positions were analyzed; smears due to complex dissociation were not included. The integrated volume for each signal was determined by measuring the identical area that encompasses the control lane with minimal background. Results of the binding assay were expressed as the relative fraction of RNA bound and plotted as a function of protein concentration using Prism 8.0 (GraphPad Software).

## Statistical methods

All error bars represent means ± SEM. Statistical analyses were performed using Prism 7.0 or 8.0 (GraphPad software). Detailed statistical methods were described in figure legends. Statistical significance: N.S., not significant.

## Reporting summary

Further information on research design is available in the Nature Portfolio Reporting Summary linked to this article.

## Data availability

All data supporting the findings of this study are available within the paper and its Supplementary Information. Major original data are deposited at the 4DN data portal (https://data.4dnucleome.org) unter the titles 4DNES9BMCFUM and 4DNESBO1XK2L. Source data are provided with this paper.

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

## Acknowledgements
We thank Drs. Bruce Armitage and Tumul Srivastava for providing the TERRA FISH probe. We thank Dr. Roger A. Greenberg (University of Pennsylvanian) Monika Golas (Universitat Augsburg), and Song Tan (The Pennsylvania State University) for kindly gifting plasmids and cell lines. We are also grateful to Dr. Kaiwei Liang (Wuhan University) for gifting pGEX-GST-His6-2_HBD plasmid and Dr. Samantha Pattenden (The University of North Carolina) for gifting Saos-2 and SK-N-FI cells We thank Dr. C. Joel McManus for critical reading of the manuscript. This work was supported by the United States National Institutes of Health (U01CA260851 to H.Z., GM118510 to D.C., GM120572 to N.J.R.).

## Author contributions
H.Z., M.X. and N.R. conceptualized this study. M.X., D.S., T.C., J.T., L.M., A.H., S.C., S.A., A.W., R.O. and A.T. designed and conducted the experiments. R. L. and D.C. designed and synthesized the dimerizers. M.X., H.Z., and R.Z. analyzed the results. M.X. made the figures. M.X., H.Z. and N.R. wrote the manuscript with comments from all authors.

## Competing interests
The authors declare no competing interests.
