## [Peer Review File · Nature Communications]

TERRA-LSD1 phase separation promotes R-loop formation for telomere maintenance in ALT cancer cellsReviewer #1 (Remarks to the Author):

To investigate the mechanism how TERRA regulates DNA synthesis on ALT telomeres, the manuscript by Xu et al. used TRF1-Fok1 system to induce strong ALT phenotypes and TFH dimerization system to recruit specific factors to telomeres in U2OS cells. With these two systems as a basis, they further applied many different sets of experimental approach in vitro and in vivo to support their conclusions, which includes fluorescence assays (immunostaining, telomere FISH, telomere DNA synthesis), FRAP, DRIP, RIP, CHIP, and in vitro phase separation assay.

Xu et al. showed that TERRA and LSD1 are mutually dependent for their telomere localization and that both knock-down of each and TERRA-interaction defective LSD1 mutants reduced ALT phenotypes in U2OS, mostly represented by less telomere clustering and less telomere DNA synthesis. They also showed that LSD1 at telomeres fused each other in U2OS cells, which is considered as an indication of LSD1 phase separation. Indeed, In vitro phase separation assay exhibited TERRA length-dependent TERRA-LSD1 condensates, which was completely lost when LSD1-TERRA interaction was disrupted by LSD1 Δ DIDR,3KE mutant. The authors also showed that TERRA-LSD1 condensates transitioned from liquid phase to solid phase (fully aggregates) when G-quadruplex structure of TERRA was disrupted by MMM. In addition, they uncovered that with TERRA-interaction defective LSD1 mutants, telomere localization of R loop was reduced in U2OS and that LSD1 enriched Rad51AP1 to telomere, which is an important factor for R loop formation. Moreover, demethylation activity of LSD1 is dispensable. Altogether, Xu et al. claimed that the formation TERRA-LSD1 condensates may concentrate R loop regulatory molecules to stimulate DNA synthesis on ALT telomeres.

This report provides lots of evident data to show that TERRA-LSD1 interaction is critical to induce ALT phenotypes and compelling data on ALT telomere regulation through TERRA-LSD1 phase separation at telomeres. So, it could represent a significant contribution to the field. However, there are a number of gaps in the experimental data and concerns with interpretation to lead the authors' conclusions.

Major issues

Roles of TERRA-LSD1 phase separation on ALT telomeres:

Throughout the manuscript, the authors tried to show that TERRA-LSD1 phase separation is produced at telomeres and argued that the formation TERRA-LSD1 condensates may regulate R loop to promote ALT telomere synthesis (Fig. 3, Fig. 4, Fig 5).

1. As the authors stated in the manuscript, there has been a previous report showing TERRA-LSD1 interaction at uncapped telomere in telomerase-positive HeLa cell and it is also independent on demethylation activity of LSD1. Then, the same TERRA-LSD1 interaction is important for both ALT telomere (U2OS) and telomerase-positive telomeres (HeLa). Is the regulation through TERRA-LSD1 phase separation ALT-specific? Or will the phase separation be also observed in HeLa cells as a common regulatory mean by TERRA-LSD1 at telomeres? The authors should investigate LSD1 phase separation at telomeres in HeLa cells by at least checking LSD1 fusion (shown as Fig. 4a and movie 1) to conclude if TERRA-LSD1 phase separation is ALT telomere-specific regulation or not.
2. The authors did not show if TERRA-LSD1 phase separation had the effects on cell growth and actual ALT telomere length. They should compare at least growth rate and telomere length by Telomere Southern between LSD1 FL (TERRA-LSD1 condensates) and LSD1 Δ DIDR,3KE (no TERRA-LSD1 condensates) in U2OS cells.
3. The authors claimed that NMM treatment transitioned TERRA-LSD1 condensates from liquid phase to solid phase (fully aggregates) by stabilizing TERRA-LSD1 interaction (Fig. 5c). Is it a general characteristic that solid phase of proteins loses their functions? If TERRA-LSD1 interaction is more stable, should it be more efficient to perform its functions such as bringing regulatory factors like Rad51AP1 to telomeres unless they are stable but displaced from telomeres? The authors also showed less telomere clustering by NMM treatment in U2OS, arguing that the

impairment of ALT machinery may be due to disrupted fluid condensates. However, the recent report showed that NMM inhibits TERRA expression (Mei et al., 2021. Sci Rep). Then, NMM-treated U2OS cells are likely to be siTERRA cells, having less TERRA and TERRA-LSD1 interaction. Therefore, the results from NMM treatment in U2OS do not directly support the author's conclusion that the impairment of ALT machinery by NMM treatment may be due to disrupted TERRA-LSD1 phase separation.

4. The authors showed that LSD1 phase separation is produced by only TERRA (G-quadruplex RNA), but not by non-G-quadruplex RNA or other tertiary RNA structure (Fig. 5 a-b), claiming that TERRA-mediated LSD1 phase separation may be specific to G-quadruplex RNA (line 368-369). However, they used only one type of G-quadruplex RNA, TERRA. They should include other G-quadruplex RNAs in addition to TERRA and the results will be more conclusive if LSD1 phase separation is G-quadruplex RNA structure specific or only specific TERRA G-quadruplex structure. Although they used RNase helicase DHX36 and poly C to disrupt G-quadruplex structure in the cells (Fig. 5) and showed less telomere clustering, those experiments cannot conclusively distinguish between G-quadruplex RNA specificity or TERRA specificity.

Recruitment of Rad51AP1 to telomeres by LSD1 and TERRA-LSD1:

1. In Fig. 6f-6h, the authors showed that Rad51-AP1 are enrichment at telomere when LSD1 was localized to telomeres and that Rad51AP1 is reduced at telomeres when LSD1 or TERRA was knockdown. So, they claimed that TERRA-LSD1 through phase separation may recruit Rad51AP1 to form R loop at telomeres and to regulate ALT telomere synthesis. However, the enrichment of Rad51AP1 at telomere was mildly increased by LSD1 and mildly reduced by siLSD1 or siTERRA. More importantly, the authors did not show any evident data if this small change of Rad51AP1 at telomeres indeed had the effects directly on R loop formation at telomeres.

Specific comments on text and figures

1. Introduction (lines 41-42, and 45). The authors repeated "these RNA". It should be more specific, describing what types of RNAs are.
2. Introduction (line 53). Dot1 is a yeast protein and METTL3, LSD1, and PRC2 are not. It needs to be noted. In addition, instead of abbreviation, the full name of Dot1 and METTL3 should be written when they were stated for the first time.
3. Introduction (line 54). The authors wrote "Through R-loop formation and interacting with various proteins". For the consistency in a sentence, it should be "Through forming R-loop and interacting with various proteins".
4. Result (lines 111-114). The authors stated that "In addition, this system allows us to induce ALT phenotypes with the addition of a small molecule 4-Hydroxyestradiol (4-OH) to translocate FokI into the nucleus, facilitating a gain or loss comparison of ALT phenotypes under different conditions." Statement appears not to be cohesive. They used TRF1-FokI system in U2OS to trigger damage exclusively within the telomeric DNA, leading to strong ALT features. However, U2OS displays ALT features as ALT cells and so there wouldn't be loss of ALT phenotypes when no further ALT activity is induced by FokI.
5. Result, Fig. 1i, j and Suppl Fig. 2a-2d (lines 124-125). Are there specific reasons for selective screening for TERRA interacting proteins with these RNA interacting proteins (NONO, hnRNPUL1, FUS, RBXP, and LSD1)? LSD1 is a known TERRA-interacting protein. What are others? In addition, instead of abbreviation, the full name of each protein except LSD1 should be written when they were stated for the first time.
6. Result (line 153). The authors stated that "Since LSD1 protects uncapped telomeres without demethylating histones". They should specify this is in telomerase-positive cells.
7. Fig. 3g-3h and Suppl. Fig. 4c-4e. +TFH should be added to figures like Fig. 3e because dimerization system was used.
8. Supp. Fig. 4e. top panel should be labeled as # of clustered telomere, not # of telomere. More importantly, siLSD1 did not show reduced telomere clustering, which is not consistent with Fig. 2b-2c.

9. Result, Fig. 3a-3b and Suppl. Fig. 4a (lines 185-189). RIP and pulldown assays with cell lysates do not indicate "direct" interaction between TERRA and LSD1.
10. Fig. 7. The authors should specify if their working model for local condensate formation at telomeres is only at G2 phase where ALT activity is active or if it is a possible model through cell cycle.
11. Method. G2 cell synchronizing (for Suppl. Fig. 1 and Suppl. Fig. 3) is missing in method.
12. Method. EMSA (for Fig. 4i and Fig. 5d) is missing in method.

Reviewer #2 (Remarks to the Author):

Xu et al. investigate the interaction between TERRA and LSD1 in the context of the ALT telomere maintenance pathway. Yet, the conclusions drawn bear an undeniable resemblance to findings previously documented, notably Porro et al. (2014), which elucidated the TERRA-LSD1 interaction, and Osterwald et al. (2015), which expounded upon the impact of LSD1 depletion on APB formation. The extensive analysis by Jia et al. (2021) into LSD1's biophysical properties further underlines the redundancy of the current research. Focusing exclusively on U2OS cells, the research appears myopic, especially when the representation of the LSD1-TERRA effect on the ALT pathway in Figure 2 lacks clarity. Additionally, given the already known capabilities of uncapped telomeres to engage TERRA-LSD1-MRE11, the impact of LSD1 depletion on the ALT pathway demands more scrutiny.

1. Redundancy with previous research and lack of novelty:

Porro et al. (2014) from Cell Reports [PMID: 24529708] already demonstrated the interaction between TERRA and LSD1. They detailed that elevated TERRA levels in TRF2-depleted cells enhance the recruitment of the LSD1-MRE11 complex at telomeres.

Osterwald et al. (2015) in Journal of Cell Science [PMID: 25908860] had previously shown that LSD1 depletion reduces APB formation.

Jia et al. (2021) in Nature Communications [PMID: 34764296] reported not only that LSD1 forms phase-separated liquid compartments but also delved into the biophysical properties and phase separation analysis of LSD1, offering a comprehensive understanding of its behavior, particularly when fused with specific proteins.

2. Limited investigation of the ALT pathway and need for a broader cell line examination:

Figure 2 falls short in providing a clear understanding of the LSD1-TERRA interaction's repercussion on the ALT pathway. The mere exploration of the TRF1-Folk1 system in U2OS cells leaves the spontaneous ALT pathway's implications shrouded. The research's limited scope, considering only U2OS cells, hampers its universality. A more comprehensive understanding would necessitate the inclusion of other cell types.

3. Limited scope of study and the questionable significance of LSD1-depletion:

Given the existing knowledge about the potential of uncapped telomere to recruit TERRA-LSD1-MRE11, the subtle impact of LSD1 depletion on the ALT pathway, especially in the long term, raises concerns.

It would be beneficial to demonstrate the results when altering the TERRA RNA sequence by changing GGG to GUG or GAG or using RNA with only three repeats of the TERRA sequence. This would help establish that the observed effects are not solely due to the TERRA sequence but are indeed a result of RNA G-quadruplex formation.

4. Image clarity:

Figures 1i and 6a need better clarity. Annotations or zoomed sections might better illustrate co-localization.

5. Inconsistency in U2OS cell presentation:

Figure 3e depicts U2OS cells that appear different. If these underwent siLSD1 treatment, it should be clarified in the main text or figure legend.

6. TERRA RNA sequence effects:

Figure 5 should include effects from altered TERRA RNA sequences or shorter RNA repeats, confirming if results stem from RNA G-quadruplex formation.

7. LSD1 phase separation with different RNA sequences:

A showcase of LSD1's behavior with various RNA sequences that form G-quadruplexes would be insightful.

8. Nomenclature error:

The legend for Figure 5 inaccurately describes N-methyl mesoporphyrin IX (NMM) as a G-quadruplex inhibitor, whereas it is a stabilizer or binding ligand.

9. Signal strength in Figure 5d:

An explanation for the stronger NMM treatment signal in comparison to just TERRA is required.

10. RAD51AP1's relation to G4-associated R-loops:

The connection between RAD51AP1 and G4-associated R-loops seems to be inadequately addressed.

Summary: this manuscript, while thorough in its approach, often revisits previously established findings, diluting its novelty. Given the substantial overlap with prior works and the limited cell types studied in relation to the ALT pathway, the current manuscript does not provide unique contributions and conceptual advance. The distinct parallels with earlier studies, combined with its narrow examination of cell types on ALT pathway, signify that this work may lack the distinctiveness and conceptual progression essential for a publication in Nature Communications, in the opinion of this reviewer.

Reviewer #3 (Remarks to the Author):

In their manuscript entitled "TERRA-LSD1 phase separation promotes R-loop formation for telomere maintenance in ALT cancer cells", Xu et al. reported that LSD1 is associated with telomere repeat-containing RNA, resulting in the emergence of telomere located phase separation and specifically affecting ALT. The work commenced with the evaluation of TERRA, a classical RNA structure that is functionally important for ALT. After screening some classic RNA-interacting proteins, the authors identified that LSD1 binds TERRA and helps localize TERRA to telomeres. Using the cellular TRF1 recruitment system, the authors showed that LSD1 plays a role in the maintenance of ALT telomeres by interacting with TERRA, independently from the enzymatic activity of LSD1. The authors demonstrated by in vitro phase separation experiments that TERRA drives LSD1 to phase separate in a length- and RNA structure-dependent manner. LSD1 condensed enriched TERRA and R-loop stimulating proteins on ALT telomeres, thereby instigating an augmentation of telomeric R-loops. Supported by a series of in vivo and in vitro experiments, their finding that LSD1 exerts a positive influence on TERRA's telomere localization and the regulation of ALT telomeres via phase separation is potentially interesting. However, some conclusions are not fully supported by the data at this stage. We feel that this manuscript could be improved if some points are evaluated more rigorously by experiments and analyses. Please see the specific points below:

Major Points

1. It seems that the depletion of TERRA led to a substantial decrease in FOKI (WT) levels in Fig. 1c and Fig. 1g. However, the reduction in FOKI levels was observed in TERRA-depleted cells in Fig. 1i. The authors might want to address the discrepancy.
2. The inconsistency regarding the change in FOKI levels upon LSD1 knockdown was also observed when comparing Fig. 2b/c and Fig. 2d.

3. According to the IP results shown in Fig. 3b, the authors concluded that the binding of TERRA by LSD1 depends on its nucleic acid binding domain but not the enzymatic activity (lines 206-208). Nonetheless, the interaction was compromised when delta-IDR was used. The binding affinity was even lower with the "delta-IDR, K661" mutant. To me, these results indicate that both the IDR and catalytic activity of LSD1 contribute to the association of TERRA. The authors could consider rewording their conclusions.

4. Again, as shown in Fig. 3b, it seems to me that both the IDR and catalytic activity of LSD1 regulate the levels of TERRA on telomere, albeit to a lesser extent than the nucleic acid binding capacity.

5. For Fig. 3g, I feel the manuscript can be improved if the authors provide higher-quality data. It is a bit annoying as the background signals varied greatly for the same type of labeling and staining. I am afraid the poor data quality could potentially interfere with the data quantification shown in 3h.

6. Have the authors tried candidates other than Rad51AP1 that may play a role in ALT? There are likely more functional candidates in this process. The readers will benefit if the authors discuss this possibility in detail in the Discussion.

Minor

1. In Fig. 4a, labeling is not aligned. Some notes are missing in Fig4c.

2. Fig5a lacks a scale.

3. Scale size is inconsistent in different figures. It would be great if authors could unify them.

4. The quality of Fig. 5c is poor. Could the authors replace them with better ones?

5. The authors might need to add a loading control in Fig. 5d.

6. The authors could provide data quantification results for all in vitro phase separation droplet formation experiments.

REVIEWER COMMENTS

Reviewer #1 (Remarks to the Author):

To investigate the mechanism how TERRA regulates DNA synthesis on ALT telomeres, the manuscript by Xu et al. used TRF1-Fok1 system to induce strong ALT phenotypes and TFH dimerization system to recruit specific factors to telomeres in U2OS cells. With these two systems as a basis, they further applied many different sets of experimental approach in vitro and in vivo to support their conclusions, which includes fluorescence assays (immunostaining, telomere FISH, telomere DNA synthesis), FRAP, DRIP, RIP, CHIP, and in vitro phase separation assay.

Xu et al. showed that TERRA and LSD1 are mutually dependent for their telomere localization and that both knock-down of each and TERRA-interaction defective LSD1 mutants reduced ALT phenotypes in U2OS, mostly represented by less telomere clustering and less telomere DNA synthesis. They also showed that LSD1 at telomeres fused each other in U2OS cells, which is considered as an indication of LSD1 phase separation. Indeed, In vitro phase separation assay exhibited TERRA length-dependent TERRA-LSD1 condensates, which was completely lost when LSD1-TERRA interaction was disrupted by LSD1 Δ DIDR,3KE mutant. The authors also showed that TERRA-LSD1 condensates transitioned from liquid phase to solid phase (fully aggregates) when G-quadruplex structure of TERRA was disrupted by MMM. In addition, they uncovered that with TERRA-interaction defective LSD1 mutants, telomere localization of R loop was reduced in U2OS and that LSD1 enriched Rad51AP1 to telomere, which is an important factor for R loop formation. Moreover, demethylation activity of LSD1 is dispensable. Altogether, Xu et al. claimed that the formation TERRA-LSD1 condensates may concentrate R loop regulatory molecules to stimulate DNA synthesis on ALT telomeres.

This report provides lots of evident data to show that TERRA-LSD1 interaction is critical to induce ALT phenotypes and compelling data on ALT telomere regulation through TERRA-LSD1 phase separation at telomeres. So, it could represent a significant contribution to the field. However, there are a number of gaps in the experimental data and concerns with interpretation to lead the authors' conclusions.

Major issues

Roles of TERRA-LSD1 phase separation on ALT telomeres:

Throughout the manuscript, the authors tried to show that TERRA-LSD1 phase separation is produced at telomeres and argued that the formation TERRA-LSD1 condensates may regulate R loop to promote ALT telomere synthesis (Fig. 3, Fig. 4, Fig 5).

1. As the authors stated in the manuscript, there has been a previous report showing TERRA-LSD1 interaction at uncapped telomere in telomerase-positive HeLa cell and it is also independent on demethylation activity of LSD1. Then, the same TERRA-LSD1

interaction is important for both ALT telomere (U2OS) and telomerase-positive telomeres (HeLa). Is the regulation through TERRA-LSD1 phase separation ALT-specific? Or will the phase separation be also observed in HeLa cells as a common regulatory mean by TERRA-LSD1 at telomeres? The authors should investigate LSD1 phase separation at telomeres in HeLa cells by at least checking LSD1 fusion (shown as Fig. 4a and movie 1) to conclude if TERRA-LSD1 phase separation is ALT telomere-specific regulation or not.

Many thanks for reviewer's comments. We investigated LSD1 phase separation at telomeres in telomerase positive Hela1.3 cells through tracking LSD1 fusion. We did not observe LSD1 phase separation in Hela cells (data added to Supplementary Fig. 5a, Movie 2). With this new data, we believe that the regulation through TERRA-LSD1 phase separation is ALT-specific. We think this can be attributed to unique features of the ALT telomeres, agreeing with previous work showing that FokI-TRF1 can induce damage at Hela telomeres but it cannot induce APB formation or telomere clustering like in ALT positive U2OS cells (Cho, 2014, Cell).

2. The authors did not show if TERRA-LSD1 phase separation had the effects on cell growth and actual ALT telomere length. They should compare at least growth rate and telomere length by Telomere Southern between LSD1 FL (TERRA-LSD1 condensates) and LSD1 Δ DIDR,3KE (no TERRA-LSD1 condensates) in U2OS cells.

We are grateful for the reviewer's suggestion. We used the colony formation assay and CellTiter-Blue assay to measure cell viability. Our data demonstrate that siLSD1 significantly reduces ALT cell growth over time. Moreover, stable overexpression of wide type LSD1 Δ IDR, which is the best control to LSD1 Δ IDR,3KE, but not DNA/RNA binding mutational LSD1 Δ IDR,3KE, could rescue the defect (data added to Fig7. c, d). In addition, telomere restriction fragment assay shows shortened telomere by LSD1 knocking down while stable LSD1 Δ IDR, but not LSD1 Δ IDR,3KE, overexpression could rescue telomere length (data added to Fig. 7a,b).

3. The authors claimed that NMM treatment transited TERRA-LSD1 condensates from liquid phase to solid phase (fully aggregates) by stabilizing TERRA-LSD1 interaction (Fig. 5c). Is it a general characteristic that solid phase of proteins loses their functions? If TERRA-LSD1 interaction is more stable, should it be more efficient to perform its functions such as bringing regulatory factors like Rad51AP1 to telomeres unless they are stable but displaced from telomeres?

Not all condensates need to be fluid to be functional. Solidification of centrosomes, for example, is thought to help sustain the spindle pulling force during cell division (Woodruff, Current Opinion in Structural Biology, 2021). Our data suggests that the function of TERRA-LSD1 is to concentrate TERRA, Rad51AP1 and other molecules

near the damaged telomere. But the proceeding of telomere repair still depends on TERRA/Rad51AP1 to interact with DNA and then to be replaced by downstream molecules. This requires the TERRA-LSD1 condensate to be fluid so TERRA/Rad51AP1 can move around to undergo transient interaction with DNA. Therefore, solidifying TERRA-LSD1 condensates would reduce their functionality, agreeing with the NMM data.

The authors also showed less telomere clustering by NMM treatment in U2OS, arguing that the impairment of ALT machinery may be due to disrupted fluid condensates. However, the recent report showed that NMM inhibits TERRA expression (Mei et al., 2021. Sci Rep). Then, NMM-treated U2OS cells are likely to be siTERRA cells, having less TERRA and TERRA-LSD1 interaction. Therefore, the results from NMM treatment in U2OS do not directly support the author's conclusion that the impairment of ALT machinery by NMM treatment may be due to disrupted TERRA-LSD1 phase separation.

We thank the reviewer for bringing our attention to the effect of NMM on TERRA expression. We investigated TERRA level in our cells using dot blot after NMM treatment for 24h. The results show that the amount of TERRA only decreased slightly by NMM treatment (please see the data in supplementary Fig. 8d). In addition, to rule out the transcription issue by NMM, we employed DHX36 and PolyC to interfere with G4 formation (Fig. 5g-j) and obtained similar defects. Combining these data, we think our results reflect disruption of TERRA-LSD1 phase separation.

4. The authors showed that LSD1 phase separation is produced by only TERRA (G-quadruplex RNA), but not by non-G-quadruplex RNA or other tertiary RNA structure (Fig. 5 a-b), claiming that TERRA-mediated LSD1 phase separation may be specific to G-quadruplex RNA (line 368-369). However, they used only one type of G-quadruplex RNA. They should include other G-quadruplex RNAs in addition to TERRA and the results will be more conclusive if LSD1 phase separation is G-quadruplex RNA structure specific or only specific TERRA G-quadruplex structure. Although they used RNase helicase DHX36 and poly C to disrupt G-quadruplex structure in the cells (Fig. 5) and showed less telomere clustering, those experiments cannot conclusively distinguish between G-quadruplex RNA specificity or TERRA specificity.

We followed the review's suggestion and added another two RNAs, LMNB and HIRA (Kharel, Nature Communications, 2023), which can form G-quadruplex as well. Our data show that those G quadruplex RNAs with different sequences can drive LSD1 phase separation. Interestingly, HIRA forms round droplets with LSD1 like TERRA, while LMNB form aggregates, maybe owing to their different interaction strengths with LSD1. Please find the data in Supplementary Fig. 9.

Recruitment of Rad51AP1 to telomeres by LSD1 and TERRA-LSD1:

1. In Fig. 6f-6h, the authors showed that Rad51-AP1 are enrichment at telomere when

LSD1 was localized to telomeres and that Rad51AP1 is reduced at telomeres when LSD1 or TERRA was knockdown. So, they claimed that TERRA-LSD1 through phase separation may recruit Rad51AP1 to form R loop at telomeres and to regulate ALT telomere synthesis.

However, the enrichment of Rad51AP1 at telomere was mildly increased by LSD1 and mildly reduced by siLSD1 or siTERRA. More importantly, the authors did not show any evident data if this small change of Rad51AP1 at telomeres indeed had the effects directly on R loop formation at telomeres.

We thank the reviewer for raising this important point. To demonstrate whether Rad51AP1 contributes to LSD1-mediated R-loop formation at telomeres, we did DRIP assay to detect R-loop level in Rad51AP1 knockout cells with LSD1 enrichment. Our data suggests that Rad51AP1 depletion decreased LSD1-induced R-loop formation, which means Rad51AP1 contributes to LSD1 function in R-loop formation (Fig. 6g). However, LSD1 enrichment in Rad51AP1 KO cells still promote R-loop formation despite at reduced level, which indicates other factors may also contribute to those elevated R loop in addition to Rad51AP1 (Fig. 6g). To identify potential factors, we plan to use ascorbate peroxidase (APEX) mediated proximity labeling assay to label proteins on telomeres and use Mass Spectrometry to detect proteins that depend on TERRA and LSD1 for localization to telomeres. Our preliminary western blot finds that LSD1, Rad51AP1, and PML reduced telomere localization after siTERRA, in agreement with data in the current manuscript. In addition, we also find other ALT proteins, including POLD3, PCNA, and Rad52, that depends on TERRA for telomere localization. Rad52 is a homologous protein that can stimulate D-loop formation and bind to R-loop (Zhang, Cell Rep. 2019; Tan, Nucleic Acids Res. 2020) and thus can potentially regulate R-loop level at ALT telomeres. Please find the data below.

In terms of the mild dependence of Rad51AP1 on LSD1, we interpret this as showing the role of TERRA-LSD1 condensates is to help the retention and not the recruitment of Rad51AP1 to telomeres. This enhancement role is typical for many condensates. A notable example would be APBs in the ALT pathway: the lack of APBs, by deletion of PML protein, slowed down telomere elongation but PML deletion cells are still able to maintain their telomere length and be viable (Loe, Genes and Development, 2020).

Specific comments on text and figures

1. Introduction (lines 41-42, and 45). The authors repeated “these RNA”. It should be more specific, describing what types of RNAs are.

We now specify them as chromatin-associated RNAs.

2. Introduction (line 53). Dot1 is a yeast protein and METTL3, LSD1, and PRC2 are not. It needs to be noted. In addition, instead of abbreviation, the full name of Dot1 and METTL3 should be written when they were stated for the first time.

We corrected the error.

3. Introduction (line 54). The authors wrote “Through R-loop formation and interacting with various proteins”. For the consistency in a sentence, it should be “Through forming R-loop and interacting with various proteins”.

We corrected the error.

4. Result (lines 111-114). The authors stated that “In addition, this system allows us to induce ALT phenotypes with the addition of a small molecule 4-Hydroxyestradiol (4-OH)

to translocate FokI into the nucleus, facilitating a gain or loss comparison of ALT phenotypes under different conditions.” Statement appears not to be cohesive. They used TRF1-FokI system in U2OS to trigger damage exclusively within the telomeric DNA, leading to strong ALT features. However, U2OS displays ALT features as ALT cells and so there wouldn't be loss of ALT phenotypes when no further ALT activity is induced by FokI.

We rewrote the sentence.

5. Result, Fig. 1i, j and Suppl Fig. 2a-2d (lines 124-125). Are there specific reasons for selective screening for TERRA interacting proteins with these RNA interacting proteins (NONO, hnRNPUL1, FUS, RBXP, and LSD1)? LSD1 is a known TERRA-interacting protein. What are others? In addition, instead of abbreviation, the full name of each protein except LSD1 should be written when they were stated for the first time.

We started with several common RNA binding proteins as first round of screening and found LSD1 and decided to follow up in this work. We are developing a proximity labeling method to identify a list of TERRA binding proteins that are specific to ALT.

6. Result (line 153). The authors stated that “Since LSD1 protects uncapped telomeres without demethylating histones”. They should specify this is in telomerase-positive cells.

We added “in telomerase-positive cells” in the sentence.

7. Fig. 3g-3h and Suppl. Fig. 4c-4e. +TFH should be added to figures like Fig. 3e because dimerization system was used.

We changed the labeling in Fig. 3g-h and Suppl. Fig. 4c-4e.

8. Supp. Fig. 4e. top panel should be labeled as # of clustered telomere, not # of telomere. More importantly, siLSD1 did not show reduced telomere clustering, which is not consistent with Fig. 2b-2c.

We thank the reviewer for catching this. Supp. Fig4e used an old method to quantify telomere clustering by counting the number of telomers as telomere clustering reduces the number of telomeres in a cell. We have switched to a new method in this paper by defining telomere clusters and count the number of clusters as described in the Method. To be consistent and avoid confusion, we re-analyzed Supp. Fig 4e with the new method. Please see the updated figure.

9. Result, Fig. 3a-3b and Suppl. Fig. 4a (lines 185-189). RIP and pulldown assays with cell lysates do not indicate "direct" interaction between TERRA and LSD1.

We have deleted “direct” so avoid over-interpretation.

10. Fig. 7. The authors should specify if their working model for local condensate formation at telomeres is only at G2 phase where ALT activity is active or if it is a possible model through cell cycle.

We limit our discussion to G2 because our current data is based on G2 where ALT is active.

11. Method. G2 cell synchronizing (for Suppl. Fig. 1 and Suppl. Fig. 3) is missing in method.

We have added the cell synchronizing protocol in Method under Cell Culture.

12. Method. EMSA (for Fig. 4i and Fig. 5d) is missing in method.

We added EMSA to the Method session.

Reviewer #2 (Remarks to the Author):

Xu et al. investigate the interaction between TERRA and LSD1 in the context of the ALT telomere maintenance pathway. Yet, the conclusions drawn bear an undeniable resemblance to findings previously documented, notably Porro et al. (2014), which elucidated the TERRA-LSD1 interaction, and Osterwald et al. (2015), which expounded upon the impact of LSD1 depletion on APB formation. The extensive analysis by Jia et al. (2021) into LSD1's biophysical properties further underlines the redundancy of the current research. Focusing exclusively on U2OS cells, the research appears myopic, especially when the representation of the LSD1-TERRA effect on the ALT pathway in Figure 2 lacks clarity. Additionally, given the already known capabilities of uncapped telomeres to engage TERRA-LSD1-MRE11, the impact of LSD1 depletion on the ALT pathway demands more scrutiny.

1. Redundancy with previous research and lack of novelty:

Porro et al. (2014) from Cell Reports [PMID: 24529708] already demonstrated the interaction between TERRA and LSD1. They detailed that elevated TERRA levels in TRF2-depleted cells enhance the recruitment of the LSD1-MRE11 complex at telomeres.

Osterwald et al. (2015) in Journal of Cell Science [PMID: 25908860] had previously shown that LSD1 depletion reduces APB formation.

Jia et al. (2021) in Nature Communications [PMID: 34764296] reported not only that LSD1 forms phase-separated liquid compartments but also delved into the biophysical properties and phase separation analysis of LSD1, offering a comprehensive understanding of its behavior, particularly when fused with specific proteins.

Compared to Porro et al, we show that TERRA and LSD1 interact in ALT cells where the telomere maintenance mechanism is different from the TRF2 depletion caused telomere stress response in the telomerase-positive cells investigated in their study. In addition, we show co-phase separation of TERRA and LSD1 that gives rise to the mutual dependence of their telomeric localization in ALT cells. Furthermore, our new data in Supplementary Fig. 5a and Movie 2 show that LSD1 does not phase separate on Hela telomeres, supporting a unique role of LSD1 in ALT cells.

Osterwald et al find LSD1 via a screen for proteins that affect APB formation. Our study adds more mechanistic understanding on how LSD1 contributes to ALT, namely through phase-separating with TERRA to enrich TERRA and R-loop stimulating proteins and promote R-loop formation.

Jia et al investigated LSD1 protein phase separation, our work focuses on RNA driven LSD1 phase separation and particularly the dependence on RNA G4 structure.

To summarize, the novelty of our work is revealing TERRA-LSD1 co-phase separation and its functional role in ALT pathway, which is not shown in any of the three studies. Our work builds on these studies but adds substantial new knowledge on several fronts and therefore we do not think our study is redundant.

2. Limited investigation of the ALT pathway and need for a broader cell line examination:

Figure 2 falls short in providing a clear understanding of the LSD1-TERRA interaction's repercussion on the ALT pathway. The mere exploration of the TRF1-Folk1 system in U2OS cells leaves the spontaneous ALT pathway's implications shrouded. The research's limited scope, considering only U2OS cells, hampers its universality. A more comprehensive understanding would necessitate the inclusion of other cell types.

We thank the reviewer for raising this important point. We added APB assay and telomere clustering assay for two more ALT+ cell lines: Soas2 and SK-N-FI. Our results suggest that knocking down LSD1 causes ALT defects in those cells, demonstrating its conserved role in ALT+ cells. Please find the data in Supplementary Fig. 3e-h.

3. Limited scope of study and the questionable significance of LSD1-depletion:
Given the existing knowledge about the potential of uncapped telomere to recruit TERRA-LSD1-MRE11, the subtle impact of LSD1 depletion on the ALT pathway, especially in the long term, raises concerns.

Thank you for raising this concern. To assess LSD1 depletion on ALT pathway in the long term, we used Telomere Restriction Fragment assay to measure telomere length after siLSD1 for 14 days and observed telomere shortening. In addition, the abundance of C-circles was also reduced, reflecting the long-term effects of siLSD1 on ALT machinery. Consistent to the molecular defects by LSD1 depletion in long term effect, cell viability is dramatically decreased, which can be partially restored by functional LSD1. Please find the data in Fig. 7 a-d.

It would be beneficial to demonstrate the results when altering the TERRA RNA sequence by changing GGG to GUG or GAG or using RNA with only three repeats of the TERRA sequence. This would help establish that the observed effects are not solely due to the TERRA sequence but are indeed a result of RNA G-quadruplex formation.

We thank the reviewer for this great suggestion. We changed the GGG sequence in the 8xTERRA to GAG and used 3xUUAGGG. We find none of them forms condensates

with LSD1, agreeing with the dependence of LSD1-TERRA phase separation on G-quadruplex. Please find the data in Supplementary Fig. 9.

4. Image clarity:

Figures 1i and 6a need better clarity. Annotations or zoomed sections might better illustrate co-localization.

We thank the reviewer for this great suggestion. We have added zoomed sections in Figure 1i and annotation arrows in Figure 6a.

5. Inconsistency in U2OS cell presentation:

Figure 3e depicts U2OS cells that appear different. If these underwent siLSD1 treatment, it should be clarified in the main text or figure legend.

In this experiment, we did not knockdown LSD1 but instead we overexpressed wide type and mutational LSD1 and used dimerization system to recruit them to telomeres. The difference in TRF1 channel reflects difference in phase separation of LSD1 mutants on telomeres. We've edited the figure caption to make it clearer.

6. TERRA RNA sequence effects:

Figure 5 should include effects from altered TERRA RNA sequences or shorter RNA repeats, confirming if results stem from RNA G-quadruplex formation.

We thank the reviewer for this great suggestion. We added altered TERRA RNA sequences (changed the GGG sequence in the 8xTERRA to GAG) and shorter RNA repeats (3xUUAGG). None of them forms condensates with LSD1, confirming that the phase separation stems from RNA G-quadruplex formation. Please find the data in Supplementary Fig. 9.

7. LSD1 phase separation with different RNA sequences:

A showcase of LSD1's behavior with various RNA sequences that form G-quadruplexes would be insightful.

We thank the reviewer for this great suggestion. We added two G-quadruplex formation RNAs, LMNB and HIRA (Kharel, Nature Communications, 2023). We found both can form condensates with LSD1, supporting the dependence on G-quadruplexes for TERRA-LSD1 phase separation. While HIRA formed round condensates like TERRA, LMNB forms irregular shaped condensates that are likely solid aggregates, suggesting different LSD1-RNA interaction can fine-tune LSD1-RNA phase separation. Please find the data in Supplementary Fig. 9.

8. Nomenclature error:

The legend for Figure 5 inaccurately describes N-methyl mesoporphyrin IX (NMM) as a G-quadruplex inhibitor, whereas it is a stabilizer or binding ligand.

We've changed the wording to state that NMM is a binding ligand.

9. Signal strength in Figure 5d:

An explanation for the stronger NMM treatment signal in comparison to just TERRA is required.

We thank the reviewer for this attention to detail. The relative stronger signal observed in NMM treated lanes likely corresponds to additional intrinsic fluorescence properties from the NMM-RNA interaction, although it does not appear to interfere with the electrophoretic mobility shift with LSD1. Rather, “NMM treatment leads to a more compact gel shift of the TERRA-LSD1 complex”. The relative difference in the intensity of the signals is one reason why gel quantitation between samples in the presence/absence of NMM is not meaningful. It appears that NMM, which preferentially binds parallel-stranded quadruplexes, may help to facilitate a more compact RNA-LSD1 complex and why we sought to include this result in the manuscript. Quadruplex-detection via NMM-based fluorescence has been extensively characterized (Kreig et. al Sua Myong, Nucleic Acids Research 2015). Our approach provided a label-free strategy to examine the LSD1-TERRA complex, which has been extensively characterized and quantitatively analyzed previously (Hirschi et. al Reiter RNA 2016; Porro et. al Lingner, Cell Reports 2014).

10. RAD51AP1's relation to G4-associated R-loops:

The connection between RAD51AP1 and G4-associated R-loops seems to be inadequately addressed.

We thank the reviewer for raising this important point. To demonstrate whether Rad51AP1 contribute to R-loop formation at telomeres, we did DRIP assay to detect R-loop level in Rad51AP1 knockout cells with LSD1 enrichment. Our data suggests that Rad51AP1 depletion decreased LSD1-induced R-loop formation, which means Rad51AP1 is necessary to LSD1 function in R-loop formation (Fig. 6g).

Summary: this manuscript, while thorough in its approach, often revisits previously established findings, diluting its novelty. Given the substantial overlap with prior works and the limited cell types studied in relation to the ALT pathway, the current manuscript does not provide unique contributions and conceptual advance. The distinct parallels with earlier studies, combined with its narrow examination of cell types on ALT pathway, signify that this work may lack the distinctiveness and conceptual progression essential for a publication in Nature Communications, in the opinion of this reviewer.

With the newly added data on 1) the effect of siSLD1 in more ALT cell line, 2) long term effect of siLSD1 on telomere length, 3) more controls for G-4 dependent LSD1 phase separation, we hope the reviewer would find the work satisfactory.

Reviewer #3 (Remarks to the Author):

In their manuscript entitled “TERRA-LSD1 phase separation promotes R-loop formation for telomere maintenance in ALT cancer cells”, Xu et al. reported that LSD1 is

associated with telomere repeat-containing RNA, resulting in the emergence of telomere located phase separation and specifically affecting ALT. The work commenced with the evaluation of TERRA, a classical RNA structure that is functionally important for ALT. After screening some classic RNA-interacting proteins, the authors identified that LSD1 binds TERRA and helps localize TERRA to telomeres. Using the cellular TRF1 recruitment system, the authors showed that LSD1 plays a role in the maintenance of ALT telomeres by interacting with TERRA, independently from the enzymatic activity of LSD1. The authors demonstrated by in vitro phase separation experiments that TERRA drives LSD1 to phase separate in a length- and RNA structure-dependent manner. LSD1 condensed enriched TERRA and R-loop stimulating proteins on ALT telomeres, thereby instigating an augmentation of telomeric R-loops. Supported by a series of in vivo and in vitro experiments, their finding that LSD1 exerts a positive influence on TERRA's telomere localization and the regulation of ALT telomeres via phase separation is potentially interesting. However, some conclusions are not fully supported by the data at this stage. We feel that this manuscript could be improved if some points are evaluated more rigorously by experiments and analyses. Please see the specific points below:

Major Points

1. It seems that the depletion of TERRA led to a substantial decrease in FOKI (WT) levels in Fig. 1c and Fig. 1g. However, the reduction in FOKI levels was observed in TERRA-depleted cells in Fig. 1i. The authors might want to address the discrepancy.

We thank the reviewer for bringing our attention to this issue. Instead of FokI level, the difference reflects changes in telomere clustering with siTERRA because FokI is fused to TRF1. That is after TERRA knocking down, telomere clustering is lost so telomeres and thus FokI foci look small and dim.

2. The inconsistency regarding the change in FOKI levels upon LSD1 knockdown was also observed when comparing Fig. 2b/c and Fig. 2d.

Like siTERRA, the difference reflects the changes in telomere clustering after LSD1 knockdown rather than FokI level.

3. According to the IP results shown in Fig. 3b, the authors concluded that the binding of TERRA by LSD1 depends on its nucleic acid binding domain but not the enzymatic activity (lines 206-208). Nonetheless, the interaction was compromised when delta-IDR was used. The binding affinity was even lower with the "delta-IDR, K661" mutant. To me, these results indicate that both the IDR and catalytic activity of LSD1 contribute to the association of TERRA. The authors could consider rewording their conclusions.

We thank the reviewer for pointing this out. We have reworded our conclusion for Figure 3b to reflect the suggested interpretation. Given LSD1^{ΔIDR, 3KE} almost abolished TERRA binding, we believe it is the dominate TERRA binding domain and our conclusions based on these mutants still stand.

4. Again, as shown in Fig. 3b, it seems to me that both the IDR and catalytic activity of LSD1 regulate the levels of TERRA on telomere, albeit to a lesser extent than the nucleic acid binding capacity.

We thank the reviewer for pointing this out. We have reworded our conclusion for Figure 3e,f to reflect this interpretation.

5. For Fig. 3g, I feel the manuscript can be improved if the authors provide higher-quality data. It is a bit annoying as the background signals varied greatly for the same type of labeling and staining. I am afraid the poor data quality could potentially interfere with the data quantification shown in 3h.

We redid the experiments to obtain higher-quality data in 3g. Please see the updated figure. We also added data points to all the bar plots to improve our figure quality.

6. Have the authors tried candidates other than Rad51AP1 that may play a role in ALT? There are likely more functional candidates in this process. The readers will benefit if the authors discuss this possibility in detail in the Discussion.

We thank the reviewer for this insightful suggestion. We plan to use ascorbate peroxidase (APEX) mediated proximity labeling assay to label proteins on telomeres and use Mass Spectrometry to detect proteins that depend on TERRA and LSD1 for localization to telomeres. Our preliminary western blot finds that LSD1, Rad51AP1, and PML reduced telomere localization after siTERRA, in agreement with data in the current manuscript. In addition, we also find other ALT proteins, including POLD3, PCNA, and Rad52, that depends on TERRA for telomere localization. Rad52 is a homologous protein that can stimulate D-loop formation and bind to R-loop (Zhang, Cell Rep. 2019; Tan, Nucleic Acids Res. 2020) and thus can potentially regulate R-loop level at ALT telomeres. Please find the data below.

Minor

1. In Fig. 4a, labeling is not aligned. Some notes are missing in Fig4c.

We corrected the error.

2. Fig5a lacks a scale.

We added the scale.

3. Scale size is inconsistent in different figures. It would be great if authors could unify them.

We made the scale size the same in same experiment.

4. The quality of Fig. 5c is poor. Could the authors replace them with better ones?

That is because the partition of LSD1 and TERRA to the solid condensates are different from the liquid droplets, giving different background signal.

5. The authors might need to add a loading control in Fig. 5d.

In contrast to the LSD1 protein, RNA is the only observable signal that is detectable with the Sybr-Safe dye. It should be noted that quantitation of Fig 5d is not possible due to intrinsic properties of NMM itself, revealing slightly stronger RNA signals relative to lanes without NMM. However, this effect does not negate the primary result of Fig5d (that "NMM treatment leads to a more compact gel shift of the TERRA-LSD1 complex").

RNA signals are observable across each of the lanes in the gel. The reviewer is correct that the protein or NMM (by themselves) could serve as loading controls although it is unclear why these additional controls would be informative. The LSD1-TERRA complex has been extensively characterized and quantitatively analyzed previously (Hirschi et. al Reiter RNA 2016 and Porro et. al Lingner, Cell Reports 2014).

6. The authors could provide data quantification results for all in vitro phase separation droplet formation experiments.

We used turbidity measurement for the droplet formation assays that need quantitative comparison. Please see Figure 5b and Supplemental Figure 9b.

Reviewer #1 (Remarks to the Author):

The authors have provided a detailed and thorough response to my questions/issues raised in my initial review, and have addressed my major concerns by adding supporting data from additional experiments/results.

I have no further concerns for the revised manuscript.

Reviewer #2 (Remarks to the Author):

The author has addressed the concerns; however, the following issues still require attention.

1. Fig 1b, 5b, S3k, and S9b, both SEM and t-test should be performed with a sample size of more than 3.
2. PFGE (Fig.7b) is not sufficiently convincing, TRF gel image lacks sufficient credibility. Therefore, it is strongly recommended for removal, as its presence does not substantially influence the conclusions drawn in this article. If the author intends to include it, it should feature at least two time points (early and late) for each experimental condition. Furthermore, it is worth noting that a 14-day inhibition period may not be sufficient to induce telomere attrition due to the end-replication problem, especially in U2OS cells, which are known to possess very long telomere lengths. (PMID: 36184605)
3. Change "Soas2" to "Saos2"
4. Add "SK-N-FI" and "Saos2" to the Material and Methods section.
5. Revise contribution for newly added authors

Reviewer #3 (Remarks to the Author):

In their revised manuscript, Xu et al. have included additional data, which address some of the previously raised concerns. However, as noted in our initial review, the manuscript still exhibits inconsistencies in data, significantly affecting the credibility of their conclusions. The authors assert that "after TERRA knocking down, telomere clustering is lost so telomeres and thus FokI foci look small and dim" in response to our first and second comments. Yet, in Figure 1I (lower panel), the size and intensity of the FokI foci in TERRA-depleted cells appear remarkably similar to those in siCtr cells, contradicting their claim. Furthermore, there are discrepancies in the number and intensity of FokI signals upon LSD1 depletion, as observed when comparing Figures 2b/d with 2i. While LSD1 knockdown seems to increase the number of FokI foci in Figures 2b/d, Figure 2i shows a decrease in FokI foci in siLSD1 cells. Additionally, the scales in several figures (Figures 4c, 4g, 4k, 4n, 5a, 5c, 6e) are too small to be discernible.

Reviewer #1 (Remarks to the Author): response in blue.

-The authors have provided a detailed and thorough response to my questions/issues raised in my initial review, and have addressed my major concerns by adding supporting data from additional experiments/results. I have no further concerns for the revised manuscript.

We are grateful to reviewer#1 for helping to strengthen the scientific rigor and conclusions of the paper.

Reviewer #2 (Remarks to the Author): response in blue.

The author has addressed the concerns; however, the following issues still require attention.

1. Fig 1b, 5b, S3k, and S9b, both SEM and t-test should be performed with a sample size of more than 3.

Many thanks for reviewer's suggestion. We performed more replicates of the experiments to ensure there are at least three independent experiments. Please refer to the new Figure 1b (page 24, line 1104), Figure 5b (page 32, line 1185), supplemental Figure 3k (page 40, line 1286), and Figure 9b (page 48, line 1380). The major conclusions of the paper are consistent with results from these additional experiments and the suggested statistical analyses.

2. PFGE (Fig.7b) is not sufficiently convincing, TRF gel image lacks sufficient credibility. Therefore, it is strongly recommended for removal, as its presence does not substantially influence the conclusions drawn in this article. If the author intends to include it, it should feature at least two time points (early and late) for each experimental condition. Furthermore, it is worth noting that a 14-day inhibition period may not be sufficient to induce telomere attrition due to the end-replication problem, especially in U2OS cells, which are known to possess very long telomere lengths. (PMID: 36184605)

We are grateful for the reviewer's suggestion. We agree that a longer time would be ideal for assessing telomere attrition. Considering the dramatic reduction in cell viability observed in siLSD1 after only 10 days (Figure. 7b in current version), it will be technically challenging to collect enough cells for telomere length analysis beyond a 14-day inhibition period. An accurate estimation of telomere 'attrition' in a longer proposed experiment (30-day inhibition period or population doubling number 25~50) may not be feasible as pointed out by the reviewer. Thus, we would like to follow the reviewer's recommendation to remove this gel from the manuscript. Please see the updated Figure 7 (page 36). As the reviewer notes, removal of this data does not diminish the main conclusions or rigor of the paper.

3. Change "Soas2" to "Saos2"

Many thanks for reviewer's correction. We have corrected the error. Please see the text (page 4, line 157 and page 41, line 1281).

4. Add "SK-N-FI" and "Saos2" to the Material and Methods section.

We thank the reviewer for pointing this out. We have added the relevant information in Material and Methods (page 12, line 562-563 and page 23, line 1078-1079).

5. Revise contribution for newly added authors

Thank you for the careful review. We have now updated author contributions (page 23, line 1087-1088).

Reviewer #3 (Remarks to the Author): response in blue.

In their revised manuscript, Xu et al. have included additional data, which address some of the previously raised concerns. However, as noted in our initial review, the manuscript still exhibits inconsistencies in data, significantly affecting the credibility of their conclusions. The authors assert that "after TERRA knocking down, telomere clustering is lost so telomeres and thus FokI foci look small and dim" in response to our first and second comments. Yet, in Figure 1I (lower panel), the size and intensity of the FokI foci in TERRA-depleted cells appear remarkably similar to those in siCtr cells, contradicting their claim.

We thank the reviewer for their attention to detail. Our conclusion of FokI foci being smaller/dimmer after TERRA knocking down is based telomere FISH results in Figure 1d because FokI is fused to TRF1. To confirm this, we quantified the number of clustered TRF1-FokI foci and the intensity of TRF1-FokI foci for data in Figure 1i and in supplemental Figure 2a-d with same method used for quantifying the FISH experiment shown in Figure 1d. Please refer to the TRF1-FokI quantification data below. Indeed, the quantification showed that siTERRA resulted in less TRF1-FokI clustering and dimer TRF1-FokI foci. There is large heterogeneity among single cells and we agree that the cells we have selected may not necessarily reflect the difference in average. To avoid confusion, we selected more representative cells to reflect the difference in average FokI size and intensity. Please see revised Fig. 1i (bottom panel, page 24). We hope that this additional quantitation of the data has alleviated the reviewer's concerns.

Furthermore, there are discrepancies in the number and intensity of FokI signals upon LSD1 depletion, as observed when comparing Figures 2b/d with 2i. While LSD1 knockdown seems to increase the number of FokI foci in Figures 2b/d, Figure 2i shows a decrease in FokI foci in siLSD1 cells.

Thank you again for noting this comment. Similar to the observed siTERRA result in Figure 1, we believe this is caused by single cell heterogeneity. To better resolve the discrepancies in the data, we quantified the number of clustered TRF1-FokI foci and the intensity of the TRF1-FokI foci for all data in Figure 2i. This was performed in an identical way as the FISH data was evaluated in Figure 2c. Please refer to the quantitation below. Agreeing with prediction from the FISH results in Figure 2c, this quantification also suggests that siLSD1 led to less TRF1-FokI foci clustering and dimmer TRF1-FokI foci. Based upon this quantification, we have also chosen the more representative cells to reflect the difference in average TRF1-FokI size and intensity. Please see the new Figure 2i (page 26).

Additionally, the scales in several figures (Figures 4c, 4g, 4k, 4n, 5a, 5c, 6e) are too small to be discernible.

We thank the reviewer for pointing this out. We have increased the size and length of scale bar in those figures. Please refer to updated figures (Figure 4c,g,k,n on page 30, Figure 5a,c on page 32, Figure 6e on page 34).